# On $f$-Divergence Principled Domain Adaptation: An Improved Framework

**Ziqiao Wang**
Tongji University
Shanghai, China
ziqiaowang@tongji.edu.cn

**Yongyi Mao**
University of Ottawa
Ottawa, Canada
ymao@uottawa.ca

## Abstract

Unsupervised domain adaptation (UDA) plays a crucial role in addressing distribution shifts in machine learning. In this work, we improve the theoretical foundations of UDA proposed in Acuna et al. (2021) by refining their $f$-divergence-based discrepancy and additionally introducing a new measure, $f$-domain discrepancy ($f$-DD). By removing the absolute value function and incorporating a scaling parameter, $f$-DD obtains novel target error and sample complexity bounds, allowing us to recover previous KL-based results and bridging the gap between algorithms and theory presented in Acuna et al. (2021). Using a localization technique, we also develop a fast-rate generalization bound. Empirical results demonstrate the superior performance of $f$-DD-based learning algorithms over previous works in popular UDA benchmarks.

## 1 Introduction

Machine learning often faces the daunting challenge of domain shift, where the distribution of data in the testing environment differs from that used in training. *Unsupervised Domain Adaptation* (UDA) arises as a solution to this problem. In UDA, models are allowed to access to labelled source domain data and unlabelled target domain data, while the ultimate goal is to find a model that performs well on the target domain. The mainstream theoretical foundations of UDA, and more broadly, domain adaptation [2], primarily rely on the seminal works of discrepancy-based theory [3, 4]. In particular, [3, 4] characterize the error gap between two domains using a hypothesis class-specified discrepancy measure e.g., $\mathcal{H}\Delta\mathcal{H}$-divergence. While these works initially focus on binary classification tasks and zero-one loss, [5] extend the theory to a more general setting. Subsequently, this kind of theoretical framework was extended by various works [6–10, 1, 11], all sharing some common properties such as the ability to estimate the proposed discrepancy from finite unlabeled samples. Importantly, these theoretical results often inspire the design of new algorithms, such as domain-adversarial training of neural networks (DANN) [12] and Margin Disparity Discrepancy (MDD) [9], directly motivated by $\mathcal{H}\Delta\mathcal{H}$-divergence.

Recently, [1] proposes an $f$-divergence-based domain learning framework, which generalizes various previous frameworks (e.g., those based on $\mathcal{H}\Delta\mathcal{H}$-divergence) and have demonstrated great empirical successes. However, we argue that this framework has potential limitations, at least in three aspects.

First, their discrepancy measure is based on the variational representation of $f$-divergence in [13] (cf. Lemma 2.1). Although this variational formula is commonly adopted, its weakness has been pointed out in several works [14, 15]. For example, from this formula, one cannot recover the well-known Donsker and Varadhan's (DV) representation of KL divergence [16]. This reveals a second limitation: some existing domain adaptation or transfer learning theories are based on the DV representation of KL, such as [17–20], and the framework by [1], although including KL as a special case of $f$-divergence, fail to unify those theories. Furthermore and more critically,

the discrepancy measure proposed by [1] contains an absolute value function, while the original variational representation does not. Notably, this absolute value function is necessary in their derivation of a target error upper bound, and the fundamental reason behind this is still due to the weak version of the variational representation relied upon. Specifically, a variational representation of an $f$-divergence is a lower bound of the divergence, but using a weak lower bound creates technical difficulties in proving an upper bound for target domain error. [1] chooses to add an absolute value function accordingly, potentially leading to an overestimation of the corresponding $f$-divergence. In fact, this absolute value function is removed in their proposed algorithm, termed $f$-Domain Adversarial Learning ($f$-DAL). While $f$-DAL outperforms both DANN and MDD in standard benchmarks, this choice exhibits a clear gap between their theory and algorithm.

In this work, to overcome these limitations and explore the full potential of $f$-divergence, we present an improved framework for $f$-divergence-based domain learning theory. Specifically, we apply a more advanced variational representation of $f$-divergence (cf. Lemma 2.2), independently developed by [21, 22] and [23]. After introducing some preliminaries, the organization of the remainder of the paper and our main contributions are summarized below.

- In Section 3, we revisit the theoretical analysis in [1], where we refine their $f$-divergence-based discrepancy by Lemma 2.2 while retaining the absolute value function in the definition. The resulting target error bound (cf. Lemma 3.1) and the KL-based generalization bound (cf. Theorem 3.1) complement the theoretical framework of [1].
- In Section 4, we design a novel $f$-divergence-based domain discrepancy measure, dubbed $f$-DD. Specifically, we eliminate the absolute value function from the definition used in Section 3, incorporating a scaling parameter instead. We then derive an upper bound for the target error (cf. Theorem 4.1) and the sample complexity bound for our $f$-DD. The generalization bound based on empirical $f$-DD naturally follows from these results. Notably, the obtained target error bound allows us to recover the previous KL-based result in [19] (cf. Corollary 4.1).
- In Section 5, to improve the convergence rate of our $f$-DD-based bound, we sharpen the bound using a localization technique [24, 10]. The localized $f$-DD allows for a crisp application of the local Rademacher complexity-based concentration results [24], while also enabling us to achieve a fast-rate target error bound (cf. Theorem 5.1). As a concrete example, we present a generalization bound based on localized KL-DD (cf. Theorem 5.2), where our proof techniques are directly connected to fast-rate PAC-Bayesian bounds [25, 26] and fast-rate information-theoretic generalization bounds [27–29].
- In Section 6, we conduct empirical studies based on our $f$-DD framework. We show $f$-DD outperforms the original $f$-DAL in three popular UDA benchmarks, with the best performance achieved by Jeffereys-DD. Additionally, we note that the training objective in $f$-DAL aligns more closely with our theory than with [1] (cf. Proposition 1). We also show that adding the absolute value function leads to overestimation and that optimizing the scaling parameter in $f$-DD may not be necessary in practical algorithms.

## 2 Preliminaries

**Notations and UDA Setup**  Let $\mathcal{X}$ and $\mathcal{Y}$ be the input space and the label space. Let $\mathcal{H}$ be the hypothesis space, where each $h \in \mathcal{H}$ is a hypothesis mapping $\mathcal{X}$ to $\mathcal{Y}$. Consider a single-source domain adaptation setting, where $\mu$ and $\nu$ are two unknown distributions on $\mathcal{X} \times \mathcal{Y}$, characterizing respectively the source domain and the target domain. Let $\mathcal{S} = \{X_i, Y_i\}_{i=1}^n \sim \mu^{\otimes n}$ be a labeled source-domain sample and $\mathcal{T} = \{X_j\}_{j=1}^m \sim \nu_{\mathcal{X}}^{\otimes m}$ be an unlabelled target-domain sample, where $\nu_{\mathcal{X}}$ is the marginal distribution of $X$ in the target domain. We use $\hat{\mu}$ and $\hat{\nu}$ to denote the empirical distributions on $\mathcal{X}$ corresponding to $\mathcal{S}$ and $\mathcal{T}$, respectively. The objective of UDA is to find a hypothesis $h \in \mathcal{H}$ based on $\mathcal{S}$ and $\mathcal{T}$ that "works well" on the target domain.

Let $\ell : \mathcal{Y} \times \mathcal{Y} \to \mathbb{R}_0^+$ be a symmetric loss (e.g., $\ell(y, y) = 0$ for all $y \in \mathcal{Y}$). The population risk for each $h \in \mathcal{H}$ in the target domain (i.e. target error) is defined as $R_\nu(h) \triangleq \mathbb{E}_{(X,Y)\sim\nu}[\ell(h(X), Y)]$, and the population risk in the source domain, $R_\mu(h)$, is defined in the same way. Since $\mu$ and $\nu$ are unknown to the learner, one often uses recourse to the empirical risk in the source domain, which, for a given $\mathcal{S}$, is defined as $R_{\hat{\mu}}(h) \triangleq \mathbb{E}_{(X,Y)\sim\hat{\mu}}[\ell(h(X), Y)] = \frac{1}{n}\sum_{i=1}^n \ell(h(X_i), Y_i)$. Furthermore, let $f_\nu$ and $f_\mu$ be the ground truth labeling functions for the target domain and source domain, respectively, i.e. $f_\nu(x) = \arg\max_{y\in\mathcal{Y}} \nu(Y = y|x)$ and $f_\mu(x) = \arg\max_{y\in\mathcal{Y}} \mu(Y = y|x)$.

With a little abuse of the notation, we will simply use $\ell(h, h')$ to represent $\ell(h(x), h'(x))$ when the same $x$ is evaluated, serving as the "disagreement" for $h$ and $h'$ on $x$. Additionally, following the conventional literature on DA theory [5], we assume that the loss function satisfies the triangle property[1]. For the readers' convenience, a summary of all notations is provided in Table 5 in the Appendix.

**Background on $f$-divergence**    The family of $f$-divergence is defined as follows.

**Definition 2.1** ($f$-divergence [30]). Let $P$ and $Q$ be two distributions on $\Theta$. Let $\phi : \mathbb{R}_+ \to \mathbb{R}$ be a convex function with $\phi(1) = 0$. If $P \ll Q$[2], then $f$-divergence is defined as $\mathrm{D}_\phi(P||Q) \triangleq \mathbb{E}_Q\left[\phi\left(\frac{dP}{dQ}\right)\right]$, where $\frac{dP}{dQ}$ is a Radon-Nikodym derivative.

The $f$-divergence family contains many popular divergences. For example, letting $\phi(x) = x \log x$ (or $x \log x + c(x - 1)$ for any constant $c$) recovers the definition of KL divergence.

The $f$-divergence discrepancy measure by [1] is motivated by the variational formula of $f$-divergence that utilizes the Legendre transformation (LT).

**Lemma 2.1** ([13]). *Let $\phi^*$ be the convex conjugate[3] of $\phi$, and $\mathcal{G} = \{g : \Theta \to \mathrm{dom}(\phi^*)\}$. Then*

$$\mathrm{D}_\phi(P||Q) = \sup_{g \in \mathcal{G}} \mathbb{E}_{\theta \sim P}\left[g(\theta)\right] - \mathbb{E}_{\theta \sim Q}\left[\phi^*(g(\theta))\right].$$

However, it is well-known that the variational formula in Lemma 2.1 does not recover the the Donsker and Varadhan's (DV) representation of KL divergence (cf. Lemma B.1). We will elaborate on this later. Recently, [23] and [21, 22] concurrently introduce a novel variational representation for $f$-divergence, which is also implicitly stated in [31, Theorem 4.2], as given below.

**Lemma 2.2** ([21, Corollary 3.5]). *The variational formula of $f$-divergence is*

$$\mathrm{D}_\phi(P||Q) = \sup_{g \in \mathcal{G}} \mathbb{E}_{\theta \sim P}\left[g(\theta)\right] - \inf_{\alpha \in \mathbb{R}} \left\{\mathbb{E}_{\theta \sim Q}\left[\phi^*(g(\theta) + \alpha)\right] - \alpha\right\}.$$

This variational representation is a "shift transformation" of Lemma 2.1 (i.e. $g \to g + \alpha$). Note that this representation shares the same optimal solution as Lemma 2.1 (clearly identified as the corresponding $f$-divergence), but Lemma 2.2 is considered tighter in the sense that the representation in Lemma 2.2 is flatter around the optimal solution. [23] provides a comprehensive study to justify this, and they also show that Lemma 2.2 can accelerate numerical estimation of $f$-divergences.

Here, to illustrate the advantage of Lemma 2.2, we use KL divergence as an example. Specifically, let $\phi(x) = x \log x - x + 1$, then its conjugate function is $\phi^*(y) = e^y - 1$. Substituting $\phi^*$ into Lemma 2.1, we have

$$\mathrm{D}_{\mathrm{KL}}(P||Q) = \sup_{g \in \mathcal{G}} \mathbb{E}_P\left[g(\theta)\right] - \mathbb{E}_Q\left[e^{g(\theta)} - 1\right]. \tag{1}$$

This representation is usually called LT-based KL. On the other hand, with the optimal $\alpha^* = -\log \mathbb{E}_Q\left[e^{g(\theta)}\right]$, Lemma 2.2 will give us

$$\mathrm{D}_{\mathrm{KL}}(P||Q) = \sup_{g \in \mathcal{G}} \mathbb{E}_P\left[g(\theta)\right] - \inf_{\alpha \in \mathbb{R}} \left\{\mathbb{E}_Q\left[e^{g(\theta)+\alpha}\right] - 1 - \alpha\right\} = \sup_{g \in \mathcal{G}} \mathbb{E}_P\left[g(\theta)\right] - \log \mathbb{E}_Q\left[e^{g(\theta)}\right]. \tag{2}$$

Notice that Eq. (2) immediately recovers the DV representation of KL. Since $\log(x) \le x - 1$ for $x > 0$, it is evident that, as a lower bound of KL divergence, Eq. (2) is pointwise tighter than Eq. (1). In Appendix C, we also show the variational representations of some other divergences obtained from Lemma 2.2.

In the context of UDA, it may be tempting to think that using a point-wise smaller quantity (in Lemma 2.1), as the key component of an upper bound for target error, is essentially desired. However,

---

[1]The triangle property of loss function indicates that $\ell(y_1, y_2) \le \ell(y_1, y_3) + \ell(y_3, y_2)$ for any $y_1, y_2, y_3 \in \mathcal{Y}$.

[2]We say that $P$ is absolutely continuous with respect to $Q$, written $P \ll Q$, if $Q(A) = 0 \implies P(A) = 0$ for all measurable sets $A \subseteq \Theta$.

[3]For a function $f : \mathcal{X} \to \mathbb{R} \cup \{-\infty, +\infty\}$, its convex conjugate is $f^*(y) \triangleq \sup_{x \in \mathrm{dom}(f)} \langle x, y \rangle - f(x)$.

neither Lemma 2.1 nor Lemma 2.2 is able to directly give such an upper bound. To elaborate, as $\phi^*(x) \geq x$ when $\phi(1) = 0$ (cf. Lemma B.2), both Lemma 2.1 and Lemma 2.2 imply that $\mathrm{D}_\phi(P||Q) \leq \sup_g \mathbb{E}_P\left[g(\theta)\right] - \mathbb{E}_Q\left[g(\theta)\right]$. Bearing this in mind, UDA typically requires an upper bound for the quantity $\sup_h \mathbb{E}_\nu\left[\ell \circ h(X)\right] - \mathbb{E}_\mu\left[\ell \circ h(X)\right]$, and simply restricting $\mathcal{G}$ in Lemma 2.1 and Lemma 2.2 to a composition of $\mathcal{H}$ and $\ell$ can only provide a lower bound for such a quantity.

Before we propose an improved discrepancy measure, we first revisit the absolute discrepancy in [1] by using Lemma 2.2 instead. This also serves as a review of the common developments in the DA theory.

## 3  Warm-Up: Refined Absolute $f$-Divergence Domain Discrepancy

Based on Lemma 2.2, we refine the discrepancy measure of [1, Definition 3] as follows.

**Definition 3.1.** For a given $h \in \mathcal{H}$, the $\widetilde{\mathrm{D}}_\phi^{h,\mathcal{H}}$ discrepancy from $\mu$ to $\nu$ is defined as

$$\widetilde{\mathrm{D}}_\phi^{h,\mathcal{H}}(\mu||\nu) \triangleq \sup_{h' \in \mathcal{H}} \left| \mathbb{E}_\mu\left[\ell(h, h')\right] - I_{\phi,\nu}^h(\ell \circ h') \right|,$$

where $I_{\phi,\nu}^h(\ell \circ h') = \inf_{\alpha \in \mathbb{R}} \{\mathbb{E}_\nu\left[\phi^*(\ell(h, h') + \alpha)\right] - \alpha\}$.

**Remark 3.1.** *Removing the absolute value function in $\widetilde{\mathrm{D}}_\phi^{h,\mathcal{H}}(\mu||\nu)$ does not alter its non-negativity. To see this, consider $h' = h$. By the definitions of $\phi$ and $\phi^*$, $\inf_\alpha \phi^*(\alpha) - \alpha = \phi(1) = 0$, we have $\mathbb{E}_\mu\left[\ell(h, h')\right] = I_{\phi,\nu}^h(\ell \circ h') = 0$. Consequently, since $h$ exists in $\mathcal{H}$, $\widetilde{\mathrm{D}}_\phi^{h,\mathcal{H}}(\mu||\nu) \geq 0$ holds even without the absolute value function. In addition, due to the absolute value function, the relation between $\widetilde{\mathrm{D}}_\phi^{h,\mathcal{H}}(\mu||\nu)$ with the one in [1] is no longer clear.*

While the absolute value function is not required for ensuring non-negativity, it is crucial for deriving the subsequent error bound for the target domain.

**Lemma 3.1.** *Let $\lambda^* = \min_{h^* \in \mathcal{H}} R_\mu(h^*) + R_\nu(h^*)$, then for any $h \in \mathcal{H}$, we have*

$$R_\nu(h) \leq R_\mu(h) + \widetilde{\mathrm{D}}_\phi^{h,\mathcal{H}}(\mu||\nu) + \lambda^*.$$

**Remark 3.2** (Regarding $\lambda^*$)**.** *This error bound shares similarities with the previous works [3, 4, 9, 10, 1]. For example, the third term $\lambda^*$ is the ideal joint risk for the DA problem. As widely discussed in prior studies, this term captures the inherent challenge in the DA task and might be inevitable [4, 32]. However, it has also been pointed out that this $\lambda^*$ term can be significantly pessimistic, particularly in the case of conditional shift [33]. In fact, similar to the $\lambda^*$-free bound in [33, Theorem 4.1], it is a simple matter to replace $\lambda^*$ with the cross-domain error term $\min\{R_\nu(f_\mu), R_\mu(f_\nu)\}$ in Lemma 3.1 (and all the other target error bounds in this paper). See Appendix D.3 for details. Given that [1] also uses $\lambda^*$, we use $\lambda^*$ in the bounds for consistency in the remainder of this paper.*

In the sequel, we will give a Rademacher complexity based generalization bound for the target error. Let $\hat{\mathfrak{R}}_S(\mathcal{F})$ denote the empirical Rademacher complexity of function class $\mathcal{F} = \{f : \mathcal{X} \to \mathbb{R}\}$ for some sample $S$ [34]. Notice that a shift transformation of a function class will not change its Rademacher complexity, so the generalization bound based on $\widetilde{\mathrm{D}}_\phi^{h,\mathcal{H}}$ closely resembles the one presented in [1, Theorem 3], which contains a Lipschitz constant of $\phi^*$. Here, we give a generalization bound specialized for $\widetilde{\mathrm{D}}_{\mathrm{KL}}^{h,\mathcal{H}}$, wherein the Lipschitz constant of $\phi^*$ can be explicitly determined.

**Theorem 3.1.** *Let $\ell(\cdot, \cdot) \in [0, 1]$. Then, for any $h \in \mathcal{H}$, with probability at least $1 - \delta$, we have*

$$R_\nu(h) \leq R_{\hat{\mu}}(h) + \widetilde{\mathrm{D}}_{\mathrm{KL}}^{h,\mathcal{H}}(\hat{\mu}||\hat{\nu}) + 2e\hat{\mathfrak{R}}_S(\mathcal{H}^\ell) + 4\hat{\mathfrak{R}}_\mathcal{T}(\mathcal{H}^\ell) + \lambda^* + \mathcal{O}\left(\sqrt{\frac{\log(1/\delta)}{n}} + \sqrt{\frac{\log(1/\delta)}{m}}\right),$$

*where $\mathcal{H}^\ell = \{x \mapsto \ell(h(x), h'(x))|h, h' \in \mathcal{H}\}$.*

[1] also applies Rademacher complexity-based bound to further upper bound $\lambda^*$ by its empirical version, namely $\hat{\lambda}^* = \min_{h^* \in \mathcal{H}} R_{\hat{\mu}}(h^*) + R_{\hat{\nu}}(h^*)$. However, since there is no target label available, $R_{\hat{\nu}}(h^*)$ is still not computable, invoking $\hat{\lambda}^*$ here has no clear advantage.

## 4 New $f$-Divergence-Based DA Theory

While $\widetilde{\mathrm{D}}_\phi^{h,\mathcal{H}}$ serves as a valid domain discrepancy measure in DA theory, it exaggerates the domain difference without appropriate control. It's noteworthy that [1] attempts to demonstrate their discrepancy with the absolute value function is upper bounded by $\mathrm{D}_\phi$ [1, Lemma 1], but this is problematic; as $\sup U \geq 0$ does not imply $\sup U = \sup |U|$ when $U$ is not a positive function. Note that the error in [1, Lemma 1] is also identified in [35]. Furthermore, when designing their $f$-DAL algorithm, they drop the absolute value function in their hypothesis-specified $f$-divergence (see Eq. (5)). Consequently, the remarkable performance of $f$-DAL reveals a significant gap from their theoretical foundation.

To bridge this gap, we introduce a new hypothesis-specific $f$-divergence-based DA framework. Our new discrepancy measure is dubbed $f$-domain discrepancy, or $f$-DD, defined without the absolute value function and with an affine transformation.

**Definition 4.1** ($f$-DD). For a given $h \in \mathcal{H}$, the $f$-DD measure $\mathrm{D}_\phi^{h,\mathcal{H}}$ is defined as

$$\mathrm{D}_\phi^{h,\mathcal{H}}(\nu||\mu) \triangleq \sup_{h' \in \mathcal{H}, t \in \mathbb{R}} \mathbb{E}_\nu\left[t \cdot \ell(h,h')\right] - I_{\phi,\mu}^h(t\ell \circ h'),$$

where $I_{\phi,\mu}^h(t\ell \circ h') = \inf_\alpha \left\{\mathbb{E}_\mu\left[\phi^*(t \cdot \ell(h,h') + \alpha)\right] - \alpha\right\}$.

**Remark 4.1.** *If $\mathcal{H}^\ell = \mathcal{G}$, then $\mathrm{D}_\phi^{h,\mathcal{H}}(\nu||\mu)$ is an affine transformation of Lemma 2.1 (i.e. $g \to tg + \alpha$) and a scaling transformation of Lemma 2.2. Importantly, unlike $\widetilde{\mathrm{D}}_\phi^{h,\mathcal{H}}$, it's easy to see that $\mathrm{D}_\phi^{h,\mathcal{H}}(\nu||\mu) \leq \mathrm{D}_\phi(\nu||\mu)$.*

Our $\mathrm{D}_\phi^{h,\mathcal{H}}(\nu||\mu)$ retains some common properties of the discrepancies defined in the DA theory literature. First, as $t = 0$ leads to $\mathbb{E}_\nu\left[t \cdot \ell(h,h')\right] = I_{\phi,\mu}^h(t\ell \circ h') = 0$, the non-negativity of $\mathrm{D}_\phi^{h,\mathcal{H}}(\nu||\mu)$ is immediately justified. In addition, its asymmetric property is also preferred in DA, as discussed in the previous works [9, 10]. Moreover, when $\mu = \nu$, by the definition of $\phi^*$, we have $\mathrm{D}_\phi^{h,\mathcal{H}}(\nu||\mu) = 0$.

To present an error bound, the routine development, as in Lemma 3.1, is insufficient; we require a general version of the "change of measure" inequality, as given below.

**Lemma 4.1.** *Let $\psi(x) \triangleq \phi(x+1)$, and $\psi^*$ is its convex conjugate. For any $h', h \in \mathcal{H}$ and $t \in \mathbb{R}$, define $K_{h',\mu}(t) \triangleq \inf_\alpha \mathbb{E}_\mu\left[\psi^*(t \cdot \ell(h,h') + \alpha)\right]$. Let $K_\mu(t) = \sup_{h' \in \mathcal{H}} K_{h',\mu}(t)$, then for any $h, h' \in \mathcal{H}$,*

$$K_\mu^*\left(\mathbb{E}_\nu\left[\ell(h,h')\right] - \mathbb{E}_\mu\left[\ell(h,h')\right]\right) \leq \mathrm{D}_\phi^{h,\mathcal{H}}(\nu||\mu),$$

*where $K_\mu^*$ is the convex conjugate of $K_\mu$.*

It is worth reminding that $K_\mu$ and $K_\mu^*$ both depend on $h$, although we ignore $h$ in the notations to avoid cluttering. We are now ready to give a target error bound based on our $f$-DD.

**Theorem 4.1.** *For any $h \in \mathcal{H}$, we have*

$$R_\nu(h) \leq R_\mu(h) + \inf_{t \geq 0} \frac{\mathrm{D}_\phi^{h,\mathcal{H}}(\nu||\mu) + K_\mu(t)}{t} + \lambda^*. \tag{3}$$

*Furthermore, let $\ell \in [0,1]$, if $\phi$ is twice differentiable and $\phi''$ is monotone, then*

$$R_\nu(h) \leq R_\mu(h) + \sqrt{\frac{2}{\phi''(1)}\mathrm{D}_\phi^{h,\mathcal{H}}(\nu||\mu)} + \lambda^*. \tag{4}$$

The following is an application of Eq. (4), where we consider the case of KL, namely $\mathrm{D}_{\mathrm{KL}}^{h,\mathcal{H}}(\nu||\mu)$.

**Corollary 4.1.** *Let $\ell \in [0,1]$, then for any $h \in \mathcal{H}$, we have $R_\nu(h) \leq R_\mu(h) + \sqrt{2\mathrm{D}_{\mathrm{KL}}^{h,\mathcal{H}}(\nu||\mu)} + \lambda^*$.*

As $\mathrm{D}_{\mathrm{KL}}^{h,\mathcal{H}}(\nu||\mu) \leq \mathrm{D}_{\mathrm{KL}}(\nu||\mu)$, the bound in [19, Theorem 4.2] can be recovered by Corollary 4.1 for the same bounded loss function. We also remark that the boundedness assumption in Corollary 4.1 can be further relaxed by applying the same sub-Gaussian assumption as in [19, Theorem 4.2].

To give a generalization bound, the next step involves obtaining a concentration result for $f$-DD, as given below.

**Lemma 4.2.** *Let $\ell \in [0, 1]$ and let $t_0$ be the optimal $t$ achieving the superum in $\mathrm{D}_\phi^{h,\mathcal{H}}(\nu||\mu)$. Assume $\phi^*$ is $L$-Lipschitz, then for any given $h$, with probability at least $1 - \delta$, we have*

$$\mathrm{D}_\phi^{h,\mathcal{H}}(\nu||\mu) \leq \mathrm{D}_\phi^{h,\mathcal{H}}(\hat{\nu}||\hat{\mu}) + 2\,|t_0|\,\hat{\mathfrak{R}}_\mathcal{T}(\mathcal{H}^\ell) + 2L\,|t_0|\,\hat{\mathfrak{R}}_\mathcal{S}(\mathcal{H}^\ell) + \mathcal{O}\left(\sqrt{\frac{\log(1/\delta)}{n}} + \sqrt{\frac{\log(1/\delta)}{m}}\right).$$

**Remark 4.2.** *The function $\phi^*$ needs not to be globally Lipschitz continuous; it can be locally Lipschitz for a bounded domain. For example, in the case of KL, $\phi^*$ is e-Lipschitz continuous when $\ell \in [0, 1]$. Moreover, although the distribution-dependent quantity $t_0$ might not always have a closed-form expression, in the case of $\chi^2$-DD, we know that $t_0 = \frac{2\left(\mathbb{E}_\nu\left[\ell(h,h'^*)\right] - \mathbb{E}_\mu\left[\ell(h,h'^*)\right]\right)}{\mathrm{Var}_\mu(\ell(h,h'^*))}$, where $h'^*$ is the corresponding optimal hypothesis.*

It is also straightforward to obtain concentration results for $K_\mu(t) - K_{\hat{\mu}}(t)$ and $R_\mu(h) - R_{\hat{\mu}}(h)$. Substituting these results into Eq. (3) obtains the final generalization bound based on $f$-DD. Or alternatively, one can directly substitute Lemma 4.2 into Eq. (4) (See Appendix D.7). However, in this case, the empirical $f$-DD and other terms in Lemma 4.2 will feature a square root, slowing down the convergence rate. We address this limitation in the following section.

## 5 Sharper Bounds via Localization

A localization technique in DA theory is recently studied in [10]. We now incorporate it into our framework with some novel applications. First, we define a localized hypothesis space, formally referred to as the (true) Rashomon set [36–38].

**Definition 5.1** (Rashomon set). Given a data distribution $\mu$, a hypothesis space $\mathcal{H}$ and a loss function $\ell$, for a Rashomon parameter $r \geq 0$, the Rashomon set $\mathcal{H}_r$ is an $r$-level subset of $\mathcal{H}$ defined as: $\mathcal{H}_r \triangleq \{h \in \mathcal{H}|R_\mu(h) \leq r\}$.

Notice that the Rashomon set $\mathcal{H}_r$ implicitly depends on the data distribution. In this paper, we specifically define $\mathcal{H}_r$ by the source domain distribution $\mu$. Then, we define our localized $f$-DD:

**Definition 5.2** (Localized $f$-DD). For a given $h \in \mathcal{H}_{r_1}$, the localized $f$-DD from $\mu$ to $\nu$ is defined as

$$\mathrm{D}_\phi^{h,\mathcal{H}_r}(\nu||\mu) \triangleq \sup_{h' \in \mathcal{H}_r, t \geq 0} \mathbb{E}_\nu\left[t\ell(h,h')\right] - I_{\phi,\mu}^h(t\ell \circ h').$$

**Remark 5.1.** *Compared with $f$-DD, localized $f$-DD restricts $h$ to $\mathcal{H}_{r_1}$ and $h'$ to $\mathcal{H}_r$, where $r_1$ and $r$ may or may not be equal. In addition, the scaling parameter $t$ is now restricted to $\mathbb{R}_0^+$ instead of $\mathbb{R}$.*

Clearly, $\mathrm{D}_\phi^{h,\mathcal{H}_r}(\mu||\nu)$ is non-decreasing when $r$ increases, and it is upper bounded by $\mathrm{D}_\phi^{h,\mathcal{H}}(\mu||\nu)$.

As also mentioned at the end of the previous section, Eq. (4) of Theorem 4.1 (and Corollary 4.1), involves a square root function for $f$-DD, potentially indicative of a slow-rate bound (e.g., if $\mathrm{D}_\phi^{h,\mathcal{H}} \in \mathcal{O}(1/n)$, then the bound decays with $\mathcal{O}(1/\sqrt{n})$). We now show that how the localized $f$-DD achieves a fast-rate error bound.

**Theorem 5.1.** *For any $h \in \mathcal{H}_{r_1}$ and constants $C_1, C_2 \in (0, +\infty)$ satisfying $K_{h',\mu}(C_1) \leq C_1 C_2 \mathbb{E}_\mu\left[\ell(h,h')\right]$ for any $h' \in \mathcal{H}_r$, the following holds:*

$$R_\nu(h) \leq R_\mu(h) + \frac{1}{C_1}\mathrm{D}_\phi^{h,\mathcal{H}_r}(\nu||\mu) + C_2 R_\mu^r(h) + \lambda_r^*,$$

*where $\lambda_r^* = \min_{h^* \in \mathcal{H}_r} R_\mu(h^*) + R_\nu(h^*)$ and $R_\mu^r(h) = \sup_{h' \in \mathcal{H}_r} \mathbb{E}_\mu\left[\ell(h,h')\right]$.*

**Remark 5.2.** *By the triangle property, $R_\mu^r(h) \leq r + r_1$. In this case, a small $r_1$ will reduce both the first term and the third term in the bound. However, determining the optimal value for $r$ is intricate. On the one hand, we hope $r$ is small so that $\mathrm{D}_\phi^{h,\mathcal{H}_r}(\nu||\mu)$ and $R_\mu^r(h)$ are both small. On the other hand, if $r$ is too small, specifically if $r < \lambda^*$, then it's possible that $\lambda_r^* > \lambda^*$ because the optimal hypothesis minimizing the joint risk may not exist in $\mathcal{H}_r$. Additionally, if both $r_1$ and $r$ are too small, the effective space for optimizing $C_1$ and $C_2$ may also be limited. Therefore, the value of $r$ involves a complex trade-off among the three terms.*

Overall, we expect $r_1$ to be as small as possible, aligning with the principle of empirical risk minimization for the source domain in practice. We may let $r > \lambda^*$ so that the optimal hypothesis is guaranteed to exist in the Rashomon set $\mathcal{H}_r$. Furthermore, if $r + r_1$ is unavoidably large, we prefer a small $C_2$ so that $C_2 R'_\mu(h)$ is small. If $\lambda^*$ itself is negligible, we use a vanishing $r + r_1$. In this case, one can focus on minimizing $1/C_1$ while allowing $C_2$ to be large.

Combining Theorem 5.1 with Lemma 4.2 and following routine steps will obtain the generalization bound based on $f$-DD, where the local Rademacher complexity [24] will be involved. However, one may feel unsatisfied without an explicit clue for the condition $K_{h',\mu}(C_1) \leq C_1 C_2 \mathbb{E}_\mu [\ell(h, h')]$ in Theorem 5.1. In fact, exploring concentration results under this condition is a central theme in obtaining fast-rate PAC-Bayesian generalization bounds [25, 39–41, 26] and the information-theoretic generalization bounds [27–29, 42]. Building upon similar ideas from these works, we now establish a sharper generalization result for our localized KL-DD measure, where the fast-rate condition is more explicit. One key ingredient is the following result.

**Lemma 5.1.** *Let $\ell \in [0, 1]$, and let the constants $C_1 > 0$ and $C_2 \in (0, 1)$ satisfy the condition $\left(e^{C_1} - C_1 - 1\right)\left(1 - \min\{r_1 + r, 1\} + C_2^2 \min\{r_1 + r, 1\}\right) \leq C_1 C_2$. Then, for any $h \in \mathcal{H}_{r_1}$ and $h' \in \mathcal{H}_r$, we have*

$$\mathbb{E}_\nu [\ell(h, h')] \leq \inf_{C_1, C_2} \frac{\mathrm{D}_{\mathrm{KL}}^{h, \mathcal{H}_r}(\nu||\mu)}{C_1} + (1 + C_2)\mathbb{E}_\mu [\ell(h, h')].$$

**Remark 5.3.** *As an extreme case, if $r + r_1 \to 0$ (implying $\mathbb{E}_\mu [\ell(h, h')] = 0$), then let $C_2 \to 1$, the condition in the lemma indicates that $C_1 < 1.26$. Hence, the optimal bound becomes $\mathbb{E}_\nu [\ell(h, h')] \leq 0.79 \mathrm{D}_{\mathrm{KL}}^{h, \mathcal{H}_r}(\nu||\mu)$. This bound remains valid even without $r + r_1 \to 0$. It holds when the Rashomon set $H_r$ is "consistent" with a given $h$, meaning all hypotheses in $H_r$ have similar predictions to $h$ on the source domain data. As an another case, if $r + r_1 \geq 1$ and $\sup_{h'} \mathbb{E}_\mu [\ell(h, h')]$ is also large, we may prefer a small $C_2$, such as setting $C_2 = 0.1$. In this case, the condition becomes $\left(e^{C_1} - C_1 - 1\right) C_2 \leq C_1$, suggesting that $C_1 < 3.74$. This results in the optimal bound $\mathbb{E}_\nu [\ell(h, h')] - \mathbb{E}_\mu [\ell(h, h')] \leq 0.27 \mathrm{D}_{\mathrm{KL}}^{h, \mathcal{H}_r}(\nu||\mu) + 0.1\mathbb{E}_\mu [\ell(h, h')]$. The condition in Lemma 5.1 is common in many fast-rate bound literature, such as [28, Theorem 3].*

We are now in a position to give a fast-rate generalization bound for localized KL-DD.

**Theorem 5.2.** *Under the conditions in Lemma 5.1. For any $h \in \mathcal{H}_{r_1}$, with probability at least $1 - \delta$, we have*

$$R_\nu(h) \leq R_{\hat{\mu}}(h) + \frac{\mathrm{D}_{\mathrm{KL}}^{h, \mathcal{H}_r}(\hat{\nu}||\hat{\mu})}{C_1} + C_2 R_\mu^r(h) + \mathcal{O}\left(\hat{\mathfrak{R}}_\mathcal{T}\left(\mathcal{H}_r^\ell\right) + \hat{\mathfrak{R}}_\mathcal{S}\left(\mathcal{H}_{\max\{r, r_1\}}^\ell\right)\right)$$

$$+ \mathcal{O}\left(\frac{\log(1/\delta)}{n} + \frac{\log(1/\delta)}{m}\right) + \mathcal{O}\left(\sqrt{\frac{(r_1 + r)\log(1/\delta)}{n}} + \sqrt{\frac{r \log(1/\delta)}{m}}\right) + \lambda_r^*.$$

**Remark 5.4.** *Due to the non-negativity of $f$-DD, a similar generalization bound also applies to the Jeffereys divergence (or symmetrized KL divergence) [43] counterpart, which is simply the sum of KL divergence and reverse KL divergence (i.e. $\mathrm{D}_{\mathrm{KL}}(\hat{\mu}||\hat{\nu}) + \mathrm{D}_{\mathrm{KL}}(\hat{\nu}||\hat{\mu})$). Furthermore, considering the fact that $\mathrm{D}_{\mathrm{KL}}(\nu||\mu) \leq \log(1 + \chi^2(\nu||\mu)) \leq \chi^2(\nu||\mu)$ [30], one might anticipate a similar bound for $\chi^2$-DD, which we defer to Appendix D.11.*

This generalization bound suggests that when $r + r_1$ is small, not only are the first four terms (including the local Rademacher complexities) reduced, but it also causes $\mathcal{O}\left(\frac{1}{n} + \frac{1}{m}\right)$ to dominate the convergence rate of the bound. In practice, when empirical risk of the source domain is always minimized to zero (i.e. the realizable case), then $r_1$ itself may have a fast vanishing rate (e.g., $\mathcal{O}(1/n)$). In Appendix D.12, we provide a concrete example to further illustrate the superiority of localized $f$-DD.

## 6 Algorithms and Experimental Results

### 6.1 Domain Adversarial Learning Algorithm

In a practical algorithm, the hypothesis space consists of two components: the representation part, denoted as $\mathcal{H}_{\mathrm{rep}} = \{h_{\mathrm{rep}} : \mathcal{X} \to \mathcal{Z}\}$, where $\mathcal{Z}$ is the representation space (e.g., the hidden output

of a neural network), and the classification part, denoted as $\mathcal{H}_{\mathrm{cls}} = \{h_{\mathrm{cls}} : \mathcal{Z} \to \mathcal{Y}\}$. Therefore, the entire hypothesis space is given by $\mathcal{H} = \{h_{\mathrm{cls}} \circ h_{\mathrm{rep}} | h_{\mathrm{rep}} \in \mathcal{H}_{\mathrm{rep}}, h_{\mathrm{cls}} \in \mathcal{H}_{\mathrm{cls}}\}$. The training objective in $f$-DAL [1] is

$$\min_{h \in \mathcal{H}} \max_{h' \in \mathcal{H}'} R_{\hat{\mu}}(h) + \eta \tilde{d}_{\hat{\mu}, \hat{\nu}}(h, h'). \qquad (5)$$

Here, $\tilde{d}_{\hat{\mu}, \hat{\nu}}(h, h') = \mathbb{E}_{\hat{\nu}}\left[\hat{\ell}(h, h')\right] - \mathbb{E}_{\hat{\mu}}\left[\phi^*\left(\hat{\ell}(h, h')\right)\right]$, where $\eta$ is a trade-off parameter and $\hat{\ell}$ is the surrogate loss used to evaluate the disagreement between $h$ and $h'$, which may or may not be the same as $\ell$. Note that, to better align with our framework, we change the order of $\hat{\mu}$ and $\hat{\nu}$ in $\tilde{d}_{\hat{\mu}, \hat{\nu}}$ in the original $f$-DAL. This modification is minor, as in either case, its optimal value belongs to $f$-divergence (such as KL and reverse KL, $\chi^2$ and reverse $\chi^2$).

Table 1: Accuracy (%) on the Office-31 benchmark.

| Method | A → W | D → W | W → D | A → D | D → A | W → A | Avg |
|---|---|---|---|---|---|---|---|
| ResNet-50 [44] | 68.4±0.2 | 96.7±0.1 | 99.3±0.1 | 68.9±0.2 | 62.5±0.3 | 60.7±0.3 | 76.1 |
| DANN [12] | 82.0±0.4 | 96.9±0.2 | 99.1±0.1 | 79.7±0.4 | 68.2±0.4 | 67.4±0.5 | 82.2 |
| MDD [9] | 94.5±0.3 | 98.4±0.1 | 100.0±.0 | 93.5±0.2 | 74.6±0.3 | 72.2±0.1 | 88.9 |
| KL [45] | 87.9±0.4 | 99.0±0.2 | 100.0±.0 | 85.6±0.6 | 70.1±1.1 | 69.3±0.7 | 85.3 |
| $f$-DAL [1] | **95.4**±0.7 | 98.8±0.1 | 100.0±.0 | 93.8±0.4 | 74.9±1.5 | 74.2±0.5 | 89.5 |
| Ours (KL-DD) | 94.9±0.7 | 98.7±0.1 | 100.0±.0 | **95.9**±0.6 | 74.6±0.9 | 74.6±0.7 | 89.8 |
| Ours ($\chi^2$-DD) | 95.3±0.2 | 98.7±0.1 | 100.0±.0 | 95.0±0.4 | 73.7±0.5 | **75.6**±0.2 | 89.7 |
| Ours (Jeffreys-DD) | 94.9±0.7 | **99.1**±0.2 | 100.0±.0 | **95.9**±0.6 | **76.0**±0.5 | 74.6±0.7 | **90.1** |

Table 2: Accuracy (%) on the Office-Home benchmark.

| Method | Ar→Cl | Ar→Pr | Ar→Rw | Cl→Ar | Cl→Pr | Cl→Rw | Pr→Ar | Pr→Cl | Pr→Rw | Rw→Ar | Rw→Cl | Rw→Pr | Avg |
|---|---|---|---|---|---|---|---|---|---|---|---|---|---|
| ResNet-50 [44] | 34.9 | 50.0 | 58.0 | 37.4 | 41.9 | 46.2 | 38.5 | 31.2 | 60.4 | 53.9 | 41.2 | 59.9 | 46.1 |
| DANN [12] | 45.6 | 59.3 | 70.1 | 47.0 | 58.5 | 60.9 | 46.1 | 43.7 | 68.5 | 63.2 | 51.8 | 76.8 | 57.6 |
| MDD [9] | 54.9 | 73.7 | 77.8 | 60.0 | 71.4 | 71.8 | 61.2 | 53.6 | 78.1 | 72.5 | 60.2 | 82.3 | 68.1 |
| $f$-DAL [1] | 54.7 | 71.7 | 77.8 | 61.0 | 72.6 | 72.2 | 60.8 | 53.4 | 80.0 | 73.3 | 60.6 | 83.8 | 68.5 |
| Ours (KL-DD) | 55.3 | 70.8 | 78.6 | 62.6 | **73.8** | 73.6 | 62.7 | 53.4 | 80.9 | **75.2** | 61.3 | **84.2** | 69.4 |
| Ours ($\chi^2$-DD) | 55.2 | 68.9 | 79.0 | 62.3 | 73.7 | 73.4 | 62.5 | 53.6 | **81.3** | 74.8 | 61.0 | 84.1 | 69.2 |
| Ours (Jefferys-DD) | **55.5** | **74.9** | **79.5** | **64.3** | **73.8** | **73.9** | **63.9** | **54.7** | **81.3** | 75.2 | **61.6** | **84.2** | **70.2** |

Eq. (5) results in an adversarial training strategy. Specifically, the outer optimization spans the entire hypothesis space. Meanwhile, within the inner optimization, given a $h = h_{\mathrm{rep}} \circ h_{\mathrm{cls}}$, the representation component $h_{\mathrm{rep}}$ is shared for $h'$. In other words, the optimization is carried out for $h'$ in $\mathcal{H}' = \mathcal{H}_{\mathrm{cls}} \circ h_{\mathrm{rep}} \triangleq \{h_{\mathrm{cls}} \circ h_{\mathrm{rep}} | h_{\mathrm{cls}} \in \mathcal{H}_{\mathrm{cls}}\}$. The overall training framework of our $f$-DD is illustrated in Figure 2 in Appendix.

Clearly, as also discussed previously, $\max_{h'} \tilde{d}_{\hat{\mu}, \hat{\nu}}(h, h') \leq \max_{h'} \left|\tilde{d}_{\hat{\mu}, \hat{\nu}}(h, h')\right|$, which presents a clear gap between the theory and algorithms in [1]. In contrast, this training objective aligns more closely with our $\mathrm{D}_{\phi}^{h, \mathcal{H}}$. Formally, we have the following result.

**Proposition 1.** *Let* $d_{\hat{\mu}, \hat{\nu}}(h, h') = \mathbb{E}_{\hat{\nu}}\left[\hat{\ell}(h, h')\right] - I_{\phi, \hat{\mu}}^h(\hat{\ell} \circ h')$. *Assume* $\mathcal{H}$ *is sufficiently large s.t.* $\hat{\ell} : \mathcal{X} \to \mathbb{R}$, *we have* $\max_{h'} \tilde{d}_{\hat{\mu}, \hat{\nu}}(h, h') = \max_{h'} d_{\hat{\mu}, \hat{\nu}}(h, h') = \mathrm{D}_{\phi}^{h, \mathcal{H}'}(\hat{\nu} || \hat{\mu}) \leq \mathrm{D}_{\phi}(\hat{\nu} || \hat{\mu})$.

In our algorithm, we use $d_{\hat{\mu}, \hat{\nu}}(h, h')$ to replace $\tilde{d}_{\hat{\mu}, \hat{\nu}}(h, h')$ in Eq. (5). Proposition 1 implies that either the optimal $\tilde{d}_{\hat{\mu}, \hat{\nu}}(h, h')$ or the optimal $d_{\hat{\mu}, \hat{\nu}}(h, h')$ coincides with $f$-DD. Moreover, as $\hat{\ell}$ is typically unbounded in practice (e.g., cross-entropy loss), considering the unbounded nature of $t$, Proposition 1 suggests an equivalence between optimizing $\hat{\ell}(h, h')$ through $h'$ and optimizing $t\ell(h, h')$ through both $t$ and $h'$. In this sense, $f$-DAL has already considered the scaling transformation. Later on we will empirically investigate whether explicitly optimizing $t$ is necessary.

Furthermore, we highlight that the training objective in our algorithm, namely Eq. (5) with $\tilde{d}_{\hat{\mu}, \hat{\nu}}(h, h')$ replaced by $d_{\hat{\mu}, \hat{\nu}}(h, h')$, is de facto motivated by the insights from Section 5. In that section, we demonstrate that when the source domain error is small, the $f$-DD term without the square-root function provides a more accurate reflection of generalization (see, for instance, Remark 5.3). Given that the empirical risk of the source domain is always minimized during training, we expect the final hypothesis to fall within the subset of $\mathcal{H}$ (i.e. Rashomon set with $r_1$).

## 6.2 Experiments

The goals of our experiments unfold in three aspects: 1) demonstrating that utilizing the $f$-DD measure (i.e., using $d_{\hat{\mu},\hat{\nu}}(h, h')$ as the training objective) leads to superior performance on the benchmarks; 2) confirming that the absolute discrepancy (i.e. $\left|\tilde{d}_{\hat{\mu},\hat{\nu}}(h, h')\right|$) leads to a degradation in performance; 3) showing that optimizing over $t$ may be unnecessary in practical scenarios.

**Dataset** We use three benchmark datasets: 1) the Office31 dataset [46], which comprises $4,652$ images across 31 categories. This dataset includes three domains: Amazon (**A**), Webcam (**W**) and DSLR (**D**); 2) the Office-Home dataset [47], consisting of $15,500$ images distributed among four domains: Artistic images (**Ar**), Clip Art (**Cl**), Product images (**Pr**), and Real-world images (**Rw**); and 3) two Digits datasets, MNIST and USPS [48] with the associated domain adaptation tasks MNIST $\rightarrow$ USPS (**M**$\rightarrow$**U**) and USPS $\rightarrow$ MNIST (**U**$\rightarrow$**M**).

Table 3: Accuracy (%) on the Digits datasets

| Method | M→U | U→M | Avg |
|---|---|---|---|
| DANN [12] | 91.8 | 94.7 | 93.3 |
| $f$-DAL [1] | 95.3 | 97.3 | 96.3 |
| Ours (KL-DD) | 95.7 | 98.1 | 96.9 |
| Ours ($\chi^2$-DD) | 95.4 | 97.3 | 96.4 |
| Ours (Jeffereys-DD) | **95.9** | **98.3** | **97.1** |

We follow the splits and evaluation protocol established by [49], where MNIST and USPS have $60,000$ and $7,291$ training images, as well as $10,000$ and $2,007$ test images, respectively.

**Discrepancy Measures** In our algorithms, we mainly focus on three specific discrepancy measures: KL-DD, $\chi^2$-DD and the weighted Jeffereys-DD. Specifically, the weighted Jeffereys-DD is $\gamma_1 D_{KL}(\hat{\mu}||\hat{\nu}) + \gamma_2 D_{KL}(\hat{\nu}||\hat{\mu})$, where $\gamma_1$ and $\gamma_2$ are tunable hyper-parameters. We note that Jeffereys divergence is not explored in [1] while it is also an $f$-divergence with $\phi(x) = (x - 1)\log x$, and advantages of Jeffereys divergence are studied in [18, 45, 19].

**Baselines and Implementation Details** The primary baseline for our study is the preceding $f$-DAL [1]. Note that, with the exception of Digits, the results reported by [1] for $f$-DAL rely on maximum values obtained from their $\chi^2$-divergence and weighted Jensen-Shannon divergence across individual sub-tasks. In our comparison, we also include DANN [12] and MDD [9] as they may be interpreted as special cases of $f$-DAL. Furthermore, we compare the results reported by [45] on the Office-31 dataset, although denoted as "KL", their method incorporates Jeffereys divergence in their algorithms.

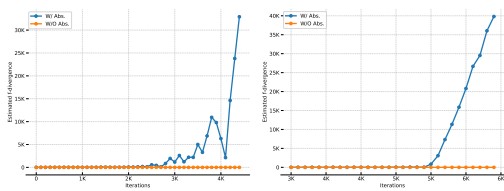

Figure 1: Failure of absolute discrepancy. The $y$-axis is the estimated $f$-divergence.

Our implementation closely follows [1]. For Office-31 and Office-Home, we utilize a pretrained ResNet-50 [44] as the backbone network, while both the primary classifier in $h$ and the auxiliary classifier in $h'$ consist of two-layer Leaky-ReLU networks. In the case of Digits, we use LeNet [50] as the backbone network and a two-layer ReLU network with Dropout (0.5) for the classifiers. Other hyper-parameter settings and the evaluation protocol remain consistent with [1], and the reported results represent average accuracies over 3 different random seeds.

**Boosted Benchmark Performance by $f$-DD**
Tables 1-3 collectively demonstrate the superior performance of our weighted Jeffereys-DD compared to other methods across the three benchmarks. Notably, the most significant improvement over $f$-DAL is observed on Office-Home (70.2% vs. 68.5%). Remarkably, this performance even surpasses the combination of $f$-DAL with a sampling-based implicit alignment

Table 4: Comparison between KL-DD and OptKL-DD

| Method | Office-31 | Office-Home | Digits |
|---|---|---|---|
| KL-DD | 89.8 | 69.4 | 96.9 |
| OptKL-DD | 89.6 | 69.2 | 96.5 |

approach [51] (See Table 6 in Appendix), specifically designed to handle the label shift issues. In addition, unlike findings in [1], where $\chi^2$ outperforms other methods on nearly all tasks, our use of a

tighter variational representation-based discrepancy reveals that $\chi^2$ is no longer superior to KL. In fact, our KL-DD slightly outperforms $\chi^2$-DD in all three benchmarks.

**Failure of Absolute Discrepancy**    We also perform experiments on **A**→**D** and **Ar**→**Cl** using the absolute discrepancy (i.e., $\max \left|\tilde{d}_{\hat{\mu},\hat{\nu}}(h,h')\right|$). Specifically, we compare the $\chi^2$-based discrepancy with (w/) and without (w/o) the absolute value function. Figure 1 illustrates that such a discrepancy can easily explode during training, demonstrating its tendency to overestimate $f$-divergence. Additional results for KL are given in Figure 3 in Appendix.

**Optimizing over** $t$    In the paragraph following Proposition 1, we discuss the observation that optimizing over $t$ may not be necessary. Empirical evidence indicates that setting $t = 1$ with hyper-parameter tuning (e.g., through $\eta$) obtains satisfactory performance. Now, let's investigate the selection of $t$ for KL-DD. Instead of using a stochastic gradient-based optimizer for updating $t$, we invoke a quadratic approximation for the optimal $t$, as studied in [23]. Specifically, we define a Gibbs measure $d\mu' = \frac{e^{\ell(h,h')}d\mu}{\mathbb{E}_\mu\left[e^{\ell(h,h')}\right]}$, then the optimal $t^* \approx 1 + \Delta t^*$, where $\Delta t^* = \frac{\mathbb{E}_\nu\left[\ell(h,h')\right] - \mathbb{E}_{\mu'}\left[\ell(h,h')\right]}{\mathrm{Var}_{\mu'}(\ell(h,h'))}$. Interested readers can find a detailed derivation of this approximation in [23, Appendix B]. Substituting $t = 1 + \Delta t^*$, we have the training objective for approximately optimal KL-DD (OptKL-DD). Table 4 presents an empirical comparison between OptKL-DD and the original KL-DD, where $t$ is simply set to 1. The results indicate that OptKL-DD does not provide any improvement on these benchmarks. Similar observations also hold for $\chi^2$, in which case optimal $t$ has an analytic form (see Appendix E), suggesting that using $t = 1$, at least for KL and $\chi^2$, might be sufficient in practice.

Additional experimental results, including an ablation study on $\eta$, t-SNE [52] visualizations and other comparisons, can be found in Appendix E.

## 7    Other Related Works

**Domain Adaptation**    Apart from those mentioned in the introduction, various other discrepancy measures are explored in DA theories and algorithms. These include the Wasserstein distance [53–55], Maximum Mean Discrepancy [56, 57], second-order statistics alignment [58, 59], transfer exponents [60, 61], Integral Probability Metrics [62] and so on. Notably, [62] defines a general Integral Probability Metrics (IPMs)-based discrepancy measure. It's noteworthy that the intersection of the IPMs family and the $f$-divergence family results in the total variation. Consequently, both corresponding discrepancy measures can consider $\mathcal{H}\Delta\mathcal{H}$-divergence as a special case. Additionally, one of our baseline models [45] diverges from the adversarial training strategy. Instead, they directly minimize the KL divergence between two isotropic Gaussian distributions (source domain Gaussian and target domain Gaussian) in the representation space. Here, the Gaussian means and variances correspond to the hidden outputs of the representation network. For further literature on DA theory, readers are directed to a recent survey by [63].

$f$**-divergence**    Moreover, the combination of $f$-divergence and adversarial training schemes has been extensively studied in generative models, including $f$-GAN [64, 65], $\chi^2$-GAN [66] and others. In the DA context, [67] introduce a $f$-divergence-based discrepancy measure while still relying on Lemma 2.1 and focusing solely on the Jensen-Shannon case. Additionally, [68] investigates $\alpha$-Rényi divergence for multi-source DA, and [69] provides some intriguing interpretations of $\chi^2$-divergence-based generalization bound for covariate shifts.

## 8    Conclusion and Future Work

In this work, we present an improved approach for integrating $f$-divergence into DA theory. Theoretical contributions include novel DA generalization bounds, including fast-rate bounds via localization. On the practical front, the revised $f$-divergence-based discrepancy improves the benchmark performance. Several promising future directions emerge from our work. Firstly, beyond its usefulness for local Rademacher complexity, the Rashomon set $\mathcal{H}_r$ also relates to another generalization measure, Rashomon ratio [37], which may give an alternative perspective on generalization in DA. Additionally, exploring transfer component-based analysis [60] for tight minimax rates in DA, invoking a power transformation instead of the affine transformation in $f$-DD, holds promise.

## Acknowledgments and Disclosure of Funding

This work is supported partly by an NSERC Discovery grant. The authors would like to thank Loïc Simon for bringing reference [35] to their attention and for the insightful feedback provided on this work. The authors also thank Fanshuang Kong for the extensive experimental guidance provided throughout this project. Furthermore, the authors are thankful to the anonymous AC and reviewers for their careful reading and valuable suggestions.

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

# Appendices

The structure of Appendix is outlined as follows: Section A provides a table summarizing the notations used throughout the paper. In Section B, we present a collection of technical lemmas crucial to our analysis. Additional variational representations for certain $f$-divergences are explored in Section C, where we also further discuss on why using a tighter variational representation of $f$-divergence is important. Section D restates our theoretical results, provides detailed proofs, and introduces supplementary theoretical findings. For more details about experiments and additional empirical results, refer to Section E.

## A  Summary of Notations

For easy reference, Table 5 summarizes the key notations used in this paper.

Table 5: Summary of notations.

| Notation | Definition |
|---|---|
| $\mathcal{X}, \mathcal{Y}, \mathcal{H}$ | input, label and hypothesis space |
| $\mu, \nu$ | source domain distribution and target domain distribution |
| $\mathcal{S}, \mathcal{T}$ | source sample $\mathcal{S} = \{(X_i, Y_i)\}_{i=1}^n$ and target sample $\mathcal{T} = \{X_i\}_{i=1}^m$ |
| $\hat{\mu}, \hat{\nu}$ | empirical source distribution and empirical target distribution |
| $\ell(h(x), h'(x))$ or $\ell(h, h')$ | loss between the predictions returned by $h$ and $h'$ |
| $R_\nu(h), R_\mu(h)$ | $R_\nu(h) \triangleq \mathbb{E}_\nu\left[\ell(h(X), Y)\right], R_\mu(h) \triangleq \mathbb{E}_\mu\left[\ell(h(X), Y)\right]$ |
| $R_{\hat{\mu}}(h)$ | $\frac{1}{n}\sum_{i=1}^n \ell(h(X), Y)$ |
| $\mathrm{D}_\phi(P\|Q)$ | $\mathbb{E}_Q\left[\phi\left(\frac{dP}{dQ}\right)\right]$; $f$-divergence between $P$ and $Q$ |
| $\widetilde{\mathrm{D}}_\phi^{h,\mathcal{H}}(\mu\|\nu)$ | $\sup_{h'\in\mathcal{H}}\left|\mathbb{E}_\mu\left[\ell(h, h')\right] - I_{\phi,\nu}^h(\ell \circ h')\right|$ |
| $\lambda^*$ | $\min_{h^*\in\mathcal{H}} R_\mu(h^*) + R_\nu(h^*)$ |
| $\hat{\mathfrak{R}}_S(\mathcal{F})$ | $\mathbb{E}_{\varepsilon_{1:n}}\left[\sup_{f\in\mathcal{F}} \frac{1}{n}\sum_{i=1}^n \varepsilon_i f(Z_i)\right]$; (empirical) Rademacher complexity |
| $\mathcal{H}^\ell$ | $\{x \mapsto \ell(h(x), h'(x))|h, h' \in \mathcal{H}\}$ |
| $\mathrm{D}_\phi^{h,\mathcal{H}}(\nu\|\mu)$ | $\sup_{h'\in\mathcal{H},t\in\mathbb{R}} \mathbb{E}_\nu\left[t \cdot \ell(h, h')\right] - I_{\phi,\mu}^h(t\ell \circ h')$; $f$-DD |
| $I_{\phi,\mu}^h(t\ell \circ h')$ | $\inf_\alpha \{\mathbb{E}_\mu\left[\phi^*(t \cdot \ell(h, h') + \alpha)\right] - \alpha\}$ |
| $\psi(x)$ | $\phi(x + 1)$ |
| $K_{h',\mu}(t)$ | $\inf_\alpha \mathbb{E}_\mu\left[\psi^*(t \cdot \ell(h, h') + \alpha)\right]$ |
| $K_\mu(t)$ | $\sup_{h'\in\mathcal{H}} K_{h',\mu}(t)$ |
| $t_0$ | optimal $t$ achieving the superum in $\mathrm{D}_\phi^{h,\mathcal{H}}(\nu\|\mu)$ |
| $\mathcal{H}_r$ | $\{h \in \mathcal{H}|R_\mu(h) \leq r\}$ |
| $\mathrm{D}_\phi^{h,\mathcal{H}_r}(\nu\|\mu)$ | $\sup_{h'\in\mathcal{H}_r,t\geq 0} \mathbb{E}_\nu\left[t\ell(h, h')\right] - I_{\phi,\mu}^h(t\ell \circ h')$ |
| $\lambda_r^*$ | $\min_{h^*\in\mathcal{H}_r} R_\mu(h^*) + R_\nu(h^*)$ |
| $R_\mu^r(h)$ | $\sup_{h'\in\mathcal{H}_r} \mathbb{E}_\mu\left[\ell(h, h')\right]$ |
| $\mathcal{H}_{\mathrm{rep}}, \mathcal{H}_{\mathrm{cls}}$ | $\{h_{\mathrm{rep}}: \mathcal{X} \to \mathcal{Z}\}$ and $\{h_{\mathrm{cls}}: \mathcal{Z} \to \mathcal{Y}\}$ |
| $\tilde{d}_{\hat{\mu},\hat{\nu}}(h, h')$ | $\mathbb{E}_{\hat{\mu}}\left[\hat{\ell}(h, h')\right] - \mathbb{E}_{\hat{\nu}}\left[\phi^*\left(\hat{\ell}(h, h')\right)\right]$ |
| $\hat{\ell}$ | surrogate loss used in practical algorithms |
| $d_{\hat{\mu},\hat{\nu}}(h, h')$ | $\mathbb{E}_{\hat{\nu}}\left[\hat{\ell}(h, h')\right] - I_{\phi,\hat{\mu}}^h(\hat{\ell} \circ h')$ |

## B  Some Technical Lemmas

The well-known Donsker-Varadhan representation of KL divergence is given below.

**Lemma B.1** (Donsker and Varadhan's variational formula). *Let $Q, P$ be probability measures on $\Theta$, for any bounded measurable function $f : \Theta \to \mathbb{R}$, we have $\mathrm{D}_{\mathrm{KL}}(Q\|P) = \sup_f \mathbb{E}_{\theta\sim Q}\left[f(\theta)\right] - \log \mathbb{E}_{\theta\sim P}\left[\exp f(\theta)\right]$.*

The following lemma is largely used.

**Lemma B.2.** *Let $\phi^*$ be the Fenchel conjugate of $\phi$ and let $\psi(x) = \phi(x+1)$, then $\psi^*(x) = \phi^*(x) - x$. Furthermore, if $\phi$ satisfies $\phi(1) = 0$, we have $\psi^*(x) \geq 0$, or equivalently $\phi^*(x) \geq x$.*

*Proof.* By definition, $\psi^*(x) = \sup_t xt - \phi(t+1)$. Let $t' = t + 1$, then $\psi^*(x) = \sup_{t'} x(t' - 1) - \phi(t') = \phi^*(x) - x$. If $\phi(1) = 0$, then $\phi^*(x) = \sup_t xt - \phi(t) \geq x - \phi(1) = x$. This completes the proof. $\square$

**Definition B.1** (Empirical Rademacher Complexity [34]). For any function class $\mathcal{F} = \{f : \mathcal{Z} \to \mathbb{R}\}$, the empirical Rademacher complexity is defined as $\hat{\mathfrak{R}}_S(\mathcal{F}) \triangleq \mathbb{E}_{\varepsilon_{1:n}} \left[ \sup_{f \in \mathcal{F}} \frac{1}{n} \sum_{i=1}^n \varepsilon_i f(Z_i) \right]$, where $S = \{Z_i\}_{i=1}^n$ and $\varepsilon_{1:n}$ is a sequence of i.i.d. Rademacher variables.

The Rademacher complexity-based generalization bound is given below.

**Lemma B.3** ([70, Theorem 3.3]). *Let $\mathcal{F}$ be a family of functions mapping from $\mathcal{Z}$ to $[0, 1]$ and let i.i.d. sample $S = \{Z_i\}_{i=1}^n$, we have $\mathbb{E}_S \left[ \sup_{f \in \mathcal{F}} \left( \mathbb{E}_Z [f(Z)] - \frac{1}{n} \sum_{i=1}^n f(Z_i) \right) \right] \leq 2 \mathbb{E}_S \left[ \hat{\mathfrak{R}}_S(\mathcal{F}) \right]$. Then for any $f \in \mathcal{F}$, with probability at least $1 - \delta$ over the draw of $S$, we have*

$$\mathbb{E}[f(Z)] \leq \frac{1}{n} \sum_{i=1}^n f(Z_i) + 2\hat{\mathfrak{R}}_S(\mathcal{F}) + \mathcal{O}\left( \sqrt{\frac{\log(1/\delta)}{n}} \right).$$

The following result is from [21, Proposition 4.1.], and the corresponding detailed proof is given in [22, Corollary 6.3.11].

**Lemma B.4.** *Assume that $\phi$ is twice differentiable on its domain and $\phi''$ is monotone. Let $\psi(x) = \phi(x + 1)$ and let $f : \mathcal{X} \to [a, b]$, then for each $t \in \mathbb{R}$, we have $\inf_\alpha \mathbb{E}_X [\psi^*(tf(X) + \alpha)] \leq \frac{(b-a)^2}{2\phi''(1)} \cdot t^2$.*

The following two lemmas are used for deriving fast-rate bound.

**Lemma B.5** ([71, Lemma 2.8]). *Let $B(x) = \frac{e^x - x - 1}{x^2}$ be the Bernstein function. If a random variable $X$ satisfies $\mathbb{E}[X] = 0$ and $X \leq b$, then $\mathbb{E}[e^X] \leq e^{B(b)\mathbb{E}[X^2]}$.*

*Proof.* It's easy to verify that $B(x)$ is an increasing function for $x > 0$. Thus, $B(x) \leq B(b)$ for $x \leq b$. Then,
$$e^x = x + 1 + x^2 B(x) \leq x + 1 + x^2 B(b).$$

For the bounded random variable $X$ with zero mean, we have

$$\mathbb{E}[e^X] \leq \mathbb{E}[X] + 1 + \mathbb{E}[X^2 B(b)] \leq e^{B(b)\mathbb{E}[X^2]}. \tag{6}$$

The last inequality is by $e^x \geq x + 1$. This completes the proof. $\square$

**Lemma B.6** (Talagrand's inequality [72, Theorem 5.4]). *Let $b > 0$ and let $\mathcal{F}$ be a class of functions from $\mathcal{X} \to \mathbb{R}$. Assume that $\sup_{f \in \mathcal{F}} \mathbb{E}[f(X)] - f(x) \leq b$ for any $x$. Then, with probability at least $1 - \delta$,*

$$\sup_{f \in \mathcal{F}} \mathbb{E}[f(X)] - \frac{1}{n} \sum_{i=1}^n f(X_i) \leq 2\mathbb{E}\left[ \sup_{f \in \mathcal{F}} \mathbb{E}[f(X)] - \frac{1}{n} \sum_{i=1}^n f(X_i) \right] + \sqrt{\frac{2\mathrm{Var}(f(X)) \log(1/\delta)}{n}} + \frac{4b \log(1/\delta)}{3n}.$$

# C Variational Representations Beyond KL Divergence

## C.1 $\chi^2$-Divergence

For $\chi^2$-divergence, let $\phi(x) = (x - 1)^2$ for $x > 0$, then $\phi^*(y) = \frac{y^2}{4} + y$.

Simply plugging $\phi^*$ into Lemma 2.1 will give us

$$\chi^2(P||Q) = \sup_g \mathbb{E}_P[g(\theta)] - \mathbb{E}_Q[g(\theta)] - \frac{\mathbb{E}_Q\left[(g(\theta))^2\right]}{4}. \tag{7}$$

Similarly, simply plugging $\phi^*$ into Lemma 2.2 will give us

$$\chi^2(P||Q) = \sup_g \mathbb{E}_P\left[g(\theta)\right] - \mathbb{E}_Q\left[g(\theta)\right] - \frac{\mathrm{Var}_Q\left(g(\theta)\right)}{4}, \tag{8}$$

where the optimal $\alpha = \mathbb{E}_Q\left[g(\theta)\right]$ in Lemma 2.2.

Notice that $\mathrm{Var}_Q\left(g(\theta)\right) \le \mathbb{E}_Q\left[\left(g(\theta)\right)^2\right]$, we can see that, as a lower bound of $\chi^2$, Eq. (8) is tighter than Eq. (7).

If we further consider the affine transformation of Lemma 2.1 and the scaling transformation of Lemma 2.2, we can recover Hammersley-Chapman-Robbins lower bound and the Cramér-Rao and van Trees lower bounds [30].

More precisely, let $g(\cdot) = ag(\cdot) + b$ be substituted to Eq. (7), where $a, b \in \mathbb{R}$, it is easy to see that

$$\chi^2(P||Q) = \sup_{g,a,b} \mathbb{E}_P\left[ag(\theta) + b\right] - \mathbb{E}_Q\left[\frac{(ag(\theta) + b)^2}{4} + ag(\theta) + b\right] = \sup_g \frac{(\mathbb{E}_P\left[g(\theta)\right] - \mathbb{E}_Q\left[g(\theta)\right])^2}{\mathrm{Var}_Q\left(g(\theta)\right)}, \tag{9}$$

where the optimal $a^* = \frac{2(\mathbb{E}_P[g(\theta)] - \mathbb{E}_Q[g(\theta)])}{\mathrm{Var}_Q(g(\theta))}$ and $b^* = -a^*\mathbb{E}_Q\left[g(\theta)\right]$.

Let $g(\cdot) = tg(\cdot)$ be substituted to Eq. (8), where $t \in \mathbb{R}$, then we have

$$\chi^2(P||Q) = \sup_{g,t} \mathbb{E}_P\left[tg(\theta)\right] - \mathbb{E}_Q\left[tg(\theta)\right] - \frac{t^2\mathrm{Var}_Q\left(g(\theta)\right)}{4} = \sup_g \frac{(\mathbb{E}_P\left[g(\theta)\right] - \mathbb{E}_Q\left[g(\theta)\right])^2}{\mathrm{Var}_Q\left(g(\theta)\right)},$$

where the optimal $t^* = a^* = \frac{2(\mathbb{E}_P[g(\theta)] - \mathbb{E}_Q[g(\theta)])}{\mathrm{Var}_Q(g(\theta))}$. Therefore, Lemma 2.2 recovers Eq. (9).

## C.2 Reverse KL Divergence

The reverse KL divergence $\mathrm{D}_{\mathrm{KL}}(Q||P)$ can be simply obtained by exchanging the orders of $P$ and $Q$ in the KL divergence $\mathrm{D}_{\mathrm{KL}}(P||Q)$. In this case, the DV representation of reverse KL is

$$\mathrm{D}_{\mathrm{KL}}(Q||P) = \sup_{g\in\mathcal{G}} \mathbb{E}_Q\left[g(\theta)\right] - \log\mathbb{E}_P\left[e^{g(\theta)}\right]. \tag{10}$$

To obtain this from Lemma 2.2, let $\phi(x) = -\log(x)$, then plugging $\phi^*(y) = -1 - \log(-y)$ into Lemma 2.2, we have

$$\mathrm{D}_{\mathrm{KL}}(Q||P) = \sup_{g,\alpha} \mathbb{E}_Q\left[g(\theta) + \alpha\right] + 1 + \mathbb{E}_P\left[\log(-g(\theta) - \alpha)\right].$$

Now reparameterizng $g + \alpha \to -e^{g+\alpha}$, we have

$$\mathrm{D}_{\mathrm{KL}}(Q||P) = \sup_{g,\alpha} -\mathbb{E}_Q\left[e^{g(\theta)+\alpha}\right] + 1 + \mathbb{E}_P\left[g(\theta) + \alpha\right] = \sup_g \mathbb{E}_Q\left[g(\theta)\right] - \log\mathbb{E}_P\left[e^{g(\theta)}\right].$$

This recovers Eq. (10).

## C.3 Jeffereys Divergence

Jeffreys divergence, a member of the $f$-divergence family with $\phi(x) = (x-1)\log x$, is the sum of KL divergence and reverse KL divergence. In our algorithm implementation, we obtain the variational formula for Jeffreys divergence by simply combining the variational representations of KL and reverse KL, namely

$$\sup_{g\in\mathcal{G}} \mathbb{E}_P\left[g(\theta)\right] - \log\mathbb{E}_Q\left[e^{g(\theta)}\right] + \sup_{g\in\mathcal{G}} \mathbb{E}_Q\left[g(\theta)\right] - \log\mathbb{E}_P\left[e^{g(\theta)}\right].$$

Moreover, there is a tight relationship between Jensen-Shannon (JS) divergence and Jeffreys divergence [73]: $\mathrm{D}_{\mathrm{JS}}(P||Q) \le \left\{\frac{1}{4}\mathrm{D}_{\mathrm{Jefferys}}(P||Q), \log\frac{2}{1+e^{-\mathrm{D}_{\mathrm{Jefferys}}(P||Q)}}\right\}$. Although we don't directly use JS divergence in our algorithms, minimizing Jeffreys divergence simultaneously minimizes JS divergence.

## C.4 Understanding the Importance of a Tighter Variational Representation

In simplified terms, Lemma 2.1 gives a variational representation expressed as:

$$\mathrm{D}_\phi(P||Q) = \sup_{g \in \mathcal{G}} V_1(g),$$

while Lemma 2.2 provides a variational representation as:

$$\mathrm{D}_\phi(P||Q) = \sup_{g \in \mathcal{G}} V_2(g).$$

where we have expressed the quantities to be maximized in Lemma 2.1 and Lemma 2.2 respectively as $V_1(g)$ and $V_2(g)$. It is clear that for each specific $g \in \mathcal{G}$, we have $V_2(g) \geq V_1(g)$ (since Lemma 2.1 is a special case of Lemma 2.2 where $\alpha = 0$). This is when we say $V_2$ is a (point-wise) tighter variational representation than $V_1$.

In the two optimization problems above, the optimal solution in both cases gives rise to same divergence. That is, when the two optimization problems can be solved perfectly, there is no advantage of one representation over the other. However, when the global optimal of the optimization problems is not attainable, any $g$ obtained in an optimization effort gives rise to an estimate of the divergence, namely as $V_1(g)$ in the first case and as $V_2(g)$ in the second. It is then clear that due to $V_2$ being a tighter variational representation, $V_2$ consistently provides a closer approximation to $\mathrm{D}_\phi(P||Q)$ than $V_1$ for the same $g$.

This virtue carries over to the min-max training strategy of UDA. In that case, $g$ is reparameterized by $t\ell \circ h'$. We desire the inner maximization, whether of $V_2$ or of $V_1$, to give rise to a good estimate of a similar divergence while the outer minimization aims to reduce this divergence. If the inner maximization is perfect, then the choice between $V_1$ and $V_2$ makes no difference. However, if the inner maximization isn't perfect, $V_2$ gives a point-wise superior result compared to $V_1$, leading to a flatter optimization region around the optimal solution. In other words, using $V_2$ as the maximization objective gives a better estimate of the corresponding $f$-divergence than $V_1$.

# D  Omitted Proofs and Additional Results

## D.1  Proof of Lemma 3.1

**Lemma 3.1.** *Let $\lambda^* = \min_{h^* \in \mathcal{H}} R_\mu(h^*) + R_\nu(h^*)$, then for any $h \in \mathcal{H}$, we have*

$$R_\nu(h) \leq R_\mu(h) + \widetilde{\mathrm{D}}_\phi^{h,\mathcal{H}}(\mu||\nu) + \lambda^*.$$

*Proof.* For a given $h \in \mathcal{H}$, let $\alpha \in \mathbb{R}$,

$$R_\nu(h) - R_\mu(h) \leq \mathbb{E}_\nu\left[\ell(h, h^*)\right] + \mathbb{E}_\nu\left[\ell(h^*, f_\nu)\right] - \mathbb{E}_\mu\left[\ell(h, f_\mu)\right] \tag{11}$$

$$= \inf_\alpha \mathbb{E}_\nu\left[\ell(h, h^*) + \alpha\right] - \alpha + \mathbb{E}_\nu\left[\ell(h^*, f_\nu)\right] - \mathbb{E}_\mu\left[\ell(h, f_\mu)\right]$$

$$\leq \inf_\alpha \mathbb{E}_\nu\left[\phi^*\left(\ell(h, h^*) + \alpha\right)\right] - \alpha + \mathbb{E}_\nu\left[\ell(h^*, f_\nu)\right] - \mathbb{E}_\mu\left[\ell(h, f_\mu)\right] \tag{12}$$

$$\leq \inf_\alpha \mathbb{E}_\nu\left[\phi^*\left(\ell(h, h^*) + \alpha\right)\right] - \alpha - \mathbb{E}_\mu\left[\ell(h, h^*)\right] + \mathbb{E}_\nu\left[\ell(h^*, f_\nu)\right] + \mathbb{E}_\mu\left[\ell(h^*, f_\mu)\right] \tag{13}$$

$$\leq \left|\inf_\alpha \mathbb{E}_\nu\left[\phi^*\left(\ell(h, h^*) + \alpha\right)\right] - \alpha - \mathbb{E}_\mu\left[\ell(h, h^*)\right]\right| + \lambda^*$$

$$\leq \widetilde{\mathrm{D}}_\phi^{h,\mathcal{H}}(\mu||\nu) + \lambda^*,$$

where Eq. (11) and (13) are by the triangle property of loss function, Eq. (12) is by Lemma B.2, and the last inequality is by the definition of $f$-DD. This completes the proof. $\square$

## D.2  Proof of Theorem 3.1

**Theorem 3.1.** *Let $\ell(\cdot, \cdot) \in [0, 1]$. Then, for any $h \in \mathcal{H}$, with probability at least $1 - \delta$, we have*

$$R_\nu(h) \leq R_{\hat{\mu}}(h) + \widetilde{\mathrm{D}}_{\mathrm{KL}}^{h,\mathcal{H}}(\hat{\mu}||\hat{\nu}) + 2e\hat{\mathfrak{R}}_{\mathcal{S}}(\mathcal{H}^\ell) + 4\hat{\mathfrak{R}}_{\mathcal{T}}(\mathcal{H}^\ell) + \lambda^* + \mathcal{O}\left(\sqrt{\frac{\log(1/\delta)}{n}} + \sqrt{\frac{\log(1/\delta)}{m}}\right),$$

*where $\mathcal{H}^\ell = \{x \mapsto \ell(h(x), h'(x)) | h, h' \in \mathcal{H}\}$.*

*Proof.* We first prove a concentration result for the $\widetilde{\mathrm{D}}_{\mathrm{KL}}^{h,\mathcal{H}}(\mu||\nu)$ measure.

**Lemma D.1.** *Assume* $\ell \in [0,1]$. *Let* $\beta$ *be a constant such that* $\beta \leq \min\left\{\mathbb{E}_\nu\left[e^{\ell(h,h')}\right], \mathbb{E}_{\hat{\nu}}\left[e^{\ell(h,h')}\right]\right\}$ *for any* $\hat{\nu}, h, h'$, *then for any given* $h$, *with probability at least* $1 - \delta$, *we have*

$$\widetilde{\mathrm{D}}_{\mathrm{KL}}^{h,\mathcal{H}}(\mu||\nu) - \widetilde{\mathrm{D}}_{\mathrm{KL}}^{h,\mathcal{H}}(\hat{\mu}||\hat{\nu}) \leq 2\hat{\mathfrak{R}}_\mathcal{S}(\mathcal{H}^\ell) + \frac{2}{\beta}\hat{\mathfrak{R}}_\mathcal{T}(\exp \circ \mathcal{H}^\ell) + \mathcal{O}\left(\sqrt{\frac{\log(1/\delta)}{n}} + \sqrt{\frac{\log(1/\delta)}{m}}\right).$$

*Proof of Lemma D.1.* For a fixed $h \in \mathcal{H}$, we have

$$\widetilde{\mathrm{D}}_{\mathrm{KL}}^{h,\mathcal{H}}(\mu||\nu) - \widetilde{\mathrm{D}}_{\mathrm{KL}}^{h,\mathcal{H}}(\hat{\mu}||\hat{\nu})$$

$$= \sup_{h'\in\mathcal{H}}\left|\mathbb{E}_\mu\left[\ell(h,h')\right] - \log\mathbb{E}_\nu\left[e^{\ell(h,h')}\right]\right| - \sup_{h'\in\mathcal{H}}\left|\mathbb{E}_{\hat{\mu}}\left[\ell(h,h')\right] - \log\mathbb{E}_{\hat{\nu}}\left[e^{\ell(h,h')}\right]\right|$$

$$\leq \sup_{h'\in\mathcal{H}}\left|\mathbb{E}_\mu\left[\ell(h,h')\right] - \log\mathbb{E}_\nu\left[e^{\ell(h,h')}\right] - \left(\mathbb{E}_{\hat{\mu}}\left[\ell(h,h')\right] - \log\mathbb{E}_{\hat{\nu}}\left[e^{\ell(h,h')}\right]\right)\right| \quad (14)$$

$$\leq \sup_{h'\in\mathcal{H}}\left|\mathbb{E}_\mu\left[\ell(h,h')\right] - \mathbb{E}_{\hat{\mu}}\left[\ell(h,h')\right]\right| + \sup_{h'\in\mathcal{H}}\left|\log\mathbb{E}_\nu\left[e^{\ell(h,h')}\right] - \log\mathbb{E}_{\hat{\nu}}\left[e^{\ell(h,h')}\right]\right|$$

$$\leq 2\hat{\mathfrak{R}}_\mathcal{S}(\mathcal{H}^\ell) + \mathcal{O}\left(\sqrt{\frac{\log(1/\delta)}{n}}\right) + \sup_{h'\in\mathcal{H}}\underbrace{\left|\log\mathbb{E}_{\hat{\nu}}\left[e^{\ell(h,h')}\right] - \log\mathbb{E}_\nu\left[e^{\ell(h,h')}\right]\right|}_{A}, \quad (15)$$

where Eq. (14) is by $\sup|U| - \sup|V| \leq \sup|U| - |V| \leq \sup||U| - |V|| \leq \sup|U - V|$ and Eq. (15) is by Lemma B.3.

Let $\beta$ be a constant such that $\beta \leq \min\left\{\mathbb{E}_\nu\left[e^{\ell(h,h')}\right], \mathbb{E}_{\hat{\nu}}\left[e^{\ell(h,h')}\right]\right\}$ for any $\hat{\nu}, h, h'$. For a given $h'$, W.L.O.G. assume that $\mathbb{E}_{\hat{\nu}}\left[e^{\ell(h,h')}\right] \geq \mathbb{E}_\nu\left[e^{\ell(h,h')}\right]$, then

$$A = \log\frac{\mathbb{E}_{\hat{\nu}}\left[e^{\ell(h,h')}\right]}{\mathbb{E}_\nu\left[e^{\ell(h,h')}\right]} = \log\left(1 + \frac{\mathbb{E}_{\hat{\nu}}\left[e^{\ell(h,h')}\right]}{\mathbb{E}_\nu\left[e^{\ell(h,h')}\right]} - 1\right)$$

$$\leq \frac{\mathbb{E}_{\hat{\nu}}\left[e^{\ell(h,h')}\right]}{\mathbb{E}_\nu\left[e^{\ell(h,h')}\right]} - 1$$

$$= \frac{1}{\mathbb{E}_\nu\left[e^{\ell(h,h')}\right]}\left(\mathbb{E}_{\hat{\nu}}\left[e^{\ell(h,h')}\right] - \mathbb{E}_\nu\left[e^{\ell(h,h')}\right]\right)$$

$$\leq \frac{1}{\beta}\left|\mathbb{E}_{\hat{\nu}}\left[e^{\ell(h,h')}\right] - \mathbb{E}_\nu\left[e^{\ell(h,h')}\right]\right|, \quad (16)$$

where the first inequality is by $\log(x + 1) \leq x$ for $x > 0$, and the last inequality is by the definition of $\beta$.

Plugging the inequality above into Eq. (15), we have

$$\widetilde{\mathrm{D}}_{\mathrm{KL}}^{h,\mathcal{H}}(\mu||\nu) - \widetilde{\mathrm{D}}_{\mathrm{KL}}^{h,\mathcal{H}}(\hat{\mu}||\hat{\nu})$$

$$\leq 2\hat{\mathfrak{R}}_\mathcal{S}(\mathcal{H}^\ell) + \mathcal{O}\left(\sqrt{\frac{\log(1/\delta)}{n}}\right) + \sup_{h'\in\mathcal{H}}\frac{1}{\beta}\left|\mathbb{E}_{\hat{\nu}}\left[e^{\ell(h,h')}\right] - \mathbb{E}_\nu\left[e^{\ell(h,h')}\right]\right|$$

$$\leq 2\hat{\mathfrak{R}}_\mathcal{S}(\mathcal{H}^\ell) + \frac{2}{\beta}\hat{\mathfrak{R}}_\mathcal{T}(\exp \circ \mathcal{H}^\ell) + \mathcal{O}\left(\sqrt{\frac{\log(1/\delta)}{n}} + \sqrt{\frac{\log(1/\delta)}{m}}\right), \quad (17)$$

where the last inequality is again by using Lemma B.3. This completes the proof. $\quad\square$

Recall Lemma 3.1, we have

$$R_\nu(h) \leq R_\mu(h) + \widetilde{D}_\phi^{h,\mathcal{H}}(\mu||\nu) + \lambda^*$$

$$\leq R_{\hat{\mu}}(h) + 2\hat{\mathfrak{R}}_\mathcal{S}(\mathcal{H}^\ell) + \mathcal{O}\left(\sqrt{\frac{\log(1/\delta)}{n}}\right) + \widetilde{D}_\phi^{h,\mathcal{H}}(\mu||\nu) + \lambda^* \tag{18}$$

$$\leq R_{\hat{\mu}}(h) + 4\hat{\mathfrak{R}}_\mathcal{S}(\mathcal{H}^\ell) + \frac{2}{\beta}\hat{\mathfrak{R}}_\mathcal{T}(\exp\circ\mathcal{H}^\ell) + \widetilde{D}_\phi^{h,\mathcal{H}}(\hat{\mu}||\hat{\nu}) + \lambda^* + \mathcal{O}\left(\sqrt{\frac{\log(1/\delta)}{n}} + \sqrt{\frac{\log(1/\delta)}{m}}\right) \tag{19}$$

$$\leq R_{\hat{\mu}}(h) + 4\hat{\mathfrak{R}}_\mathcal{S}(\mathcal{H}^\ell) + \frac{2e}{\beta}\hat{\mathfrak{R}}_\mathcal{T}(\mathcal{H}^\ell) + \widetilde{D}_\phi^{h,\mathcal{H}}(\hat{\mu}||\hat{\nu}) + \lambda^* + \mathcal{O}\left(\sqrt{\frac{\log(1/\delta)}{n}} + \sqrt{\frac{\log(1/\delta)}{m}}\right), \tag{20}$$

where Eq. (18) and Eq. (19) are by Lemma B.3 and Lemma D.1, respectively. For the last inequality, notice that the loss is bounded between $[0,1]$ so the exponential function is $e$-Lipschitz in this domain, then we apply the Talagrand's lemma [70, Lemma 5.7], i.e. $\hat{\mathfrak{R}}_\mathcal{T}(\exp\circ\mathcal{H}^\ell) \leq e\hat{\mathfrak{R}}_\mathcal{T}(\mathcal{H}^\ell)$.

Finally, due to the boundedness of the loss, there always exists a constant $\beta \in [1,e]$ such that $\beta \leq \min\left\{\mathbb{E}_\nu\left[e^{\ell(h,h')}\right], \mathbb{E}_{\hat{\nu}}\left[e^{\ell(h,h')}\right]\right\}$, and a simple choice is $\beta = 1$. Plugging $\beta = 1$ into Eq. (20) will complete the proof. $\qquad\square$

### D.3 Joint Error-Free Target Error Bound

[33] presents a $\lambda^*$-free target error bound by using the cross-domain error $\min\{R_\nu(f_\mu), R_\mu(f_\nu)\}$. We now illustrate that it is possible to incorporate this term into all of our target error bounds.

In particular, the $\lambda^*$ term appears due to the following triangle inequalities:

$$R_\nu(h) - R_\mu(h) \leq \mathbb{E}_\nu\left[\ell(h,h^*)\right] + \mathbb{E}_\nu\left[\ell(h^*,f_\nu)\right] - \mathbb{E}_\mu\left[\ell(h,f_\mu)\right]$$

$$\leq \mathbb{E}_\nu\left[\ell(h,h^*)\right] + \mathbb{E}_\nu\left[\ell(h^*,f_\nu)\right] - \mathbb{E}_\mu\left[\ell(h,h^*)\right] + \mathbb{E}_\mu\left[\ell(h^*,f_\mu)\right]$$

$$= \underbrace{\mathbb{E}_\nu\left[\ell(h,h^*)\right] - \mathbb{E}_\mu\left[\ell(h,h^*)\right]}_{A_1} + \lambda^*.$$

To avoid $\lambda^*$, we use $f_\mu$ and $f_\nu$ instead as the middle hypothesis for the application of triangle inequalities:

$$R_\nu(h) - R_\mu(h) \leq \underbrace{\mathbb{E}_\nu\left[\ell(h,f_\nu)\right] - \mathbb{E}_\mu\left[\ell(h,f_\nu)\right]}_{A_2} + \mathbb{E}_\mu\left[\ell(f_\nu,f_\mu)\right], \tag{21}$$

$$R_\nu(h) - R_\mu(h) \leq \underbrace{\mathbb{E}_\nu\left[\ell(h,f_\mu)\right] - \mathbb{E}_\mu\left[\ell(h,f_\mu)\right]}_{A_3} + \mathbb{E}_\nu\left[\ell(f_\mu,f_\nu)\right]. \tag{22}$$

If $|A_2|$ and $|A_3|$ are upper bounded by some hypothesis class-specific divergence $d_\mathcal{H}(\mu,\nu)$, then combining Eq (21) and Eq. (22) will give us:

$$R_\nu(h) \leq R_\mu(h) + d_\mathcal{H}(\mu,\nu) + \min\{R_\nu(f_\mu), R_\mu(f_\nu)\},$$

where $R_\nu(f_\mu) = \mathbb{E}_\nu\left[\ell(f_\mu,f_\nu)\right]$ and $R_\mu(f_\nu) = \mathbb{E}_\mu\left[\ell(f_\nu,f_\mu)\right]$. Clearly, if $\mathcal{H}$ is large enough such that $f_\mu$ and $f_\nu$ are all achievable, $|A_1|$, $|A_2|$ and $|A_3|$ are all upper bounded by our $f$-DD defined in this paper. In fact, as long as $f_\mu$ and $f_\nu$ are close enough to $\mathcal{H}$, our $f$-DD can upper bound $|A_2|$ and $|A_3|$, with similar adaptations used in [33, Lemma 4.1].

### D.4 Proof of Lemma 4.1

**Lemma 4.1.** *Let $\psi(x) \triangleq \phi(x+1)$, and $\psi^*$ is its convex conjugate. For any $h',h \in \mathcal{H}$ and $t \in \mathbb{R}$, define $K_{h',\mu}(t) \triangleq \inf_\alpha \mathbb{E}_\mu\left[\psi^*(t\cdot\ell(h,h') + \alpha)\right]$. Let $K_\mu(t) = \sup_{h'\in\mathcal{H}} K_{h',\mu}(t)$, then for any $h,h' \in \mathcal{H}$,*

$$K_\mu^*\left(\mathbb{E}_\nu\left[\ell(h,h')\right] - \mathbb{E}_\mu\left[\ell(h,h')\right]\right) \leq D_\phi^{h,\mathcal{H}}(\nu||\mu),$$

*where $K_\mu^*$ is the convex conjugate of $K_\mu$.*

*Proof.* By Lemma B.2, we know that

$$
\begin{aligned}
K_{h',\mu}(t) &= \inf_{\alpha} \mathbb{E}_{\mu}\left[\psi^*(t\ell(h,h') + \alpha)\right] \\
&= \inf_{\alpha} \mathbb{E}_{\mu}\left[\phi^*(t\ell(h,h') + \alpha)\right] - \alpha - t\mathbb{E}_{\mu}\left[\ell(h,h')\right] = I^h_{\phi,\mu}(t\ell \circ h') - t\mathbb{E}_{\mu}\left[\ell(h,h')\right].
\end{aligned}
$$

According to the definition of $K_{\mu}$, for each $t$, we have

$$
K_{h',\mu}(t) = I^h_{\phi,\mu}(t\ell \circ h') - t\mathbb{E}_{\mu}\left[\ell(h,h')\right] \leq K_{\mu}(t).
$$

This indicates that

$$
t\mathbb{E}_{\nu}\left[\ell(h,h')\right] - t\mathbb{E}_{\nu}\left[\ell(h,h')\right] + I^h_{\phi,\mu}(t\ell \circ h') - t\mathbb{E}_{\mu}\left[\ell(h,h')\right] \leq K_{\mu}(t).
$$

Since this inequality holds for any $t$, by rearranging terms, we have

$$
\sup_t t\left(\mathbb{E}_{\nu}\left[\ell(h,h')\right] - \mathbb{E}_{\mu}\left[\ell(h,h')\right]\right) - K_{\mu}(t) \leq \sup_t t\mathbb{E}_{\nu}\left[\ell(h,h')\right] - I^h_{\phi,\mu}(t\ell \circ h') \leq \mathrm{D}^{h,\mathcal{H}}_{\phi}(\nu\|\mu).
\tag{23}
$$

Notice that the most left hand side is the definition of $K^*_{\mu}$, we thus have

$$
K^*_{\mu}\left(\mathbb{E}_{\nu}\left[\ell(h,h')\right] - \mathbb{E}_{\mu}\left[\ell(h,h')\right]\right) \leq \mathrm{D}^{h,\mathcal{H}}_{\phi}(\nu\|\mu).
$$

This completes the proof. $\qquad\square$

## D.5 Proof of Theorem 4.1

**Theorem 4.1.** *For any $h \in \mathcal{H}$, we have*

$$
R_{\nu}(h) \leq R_{\mu}(h) + \inf_{t \geq 0} \frac{\mathrm{D}^{h,\mathcal{H}}_{\phi}(\nu\|\mu) + K_{\mu}(t)}{t} + \lambda^*.
\tag{3}
$$

*Furthermore, let $\ell \in [0,1]$, if $\phi$ is twice differentiable and $\phi''$ is monotone, then*

$$
R_{\nu}(h) \leq R_{\mu}(h) + \sqrt{\frac{2}{\phi''(1)}\mathrm{D}^{h,\mathcal{H}}_{\phi}(\nu\|\mu)} + \lambda^*.
\tag{4}
$$

*Proof.* We first follow the similar developments in Lemma 3.1.

$$
\begin{aligned}
R_{\nu}(h) - R_{\mu}(h) &\leq \mathbb{E}_{\nu}\left[\ell(h,h^*)\right] + \mathbb{E}_{\nu}\left[\ell(h^*,f_{\nu})\right] - \mathbb{E}_{\mu}\left[\ell(h,f_{\mu})\right] \\
&\leq \mathbb{E}_{\nu}\left[\ell(h,h^*)\right] + \mathbb{E}_{\nu}\left[\ell(h^*,f_{\nu})\right] - \mathbb{E}_{\mu}\left[\ell(h,h^*)\right] + \mathbb{E}_{\mu}\left[\ell(h^*,f_{\mu})\right] \\
&= \underbrace{\mathbb{E}_{\nu}\left[\ell(h,h^*)\right] - \mathbb{E}_{\mu}\left[\ell(h,h^*)\right]}_{A} + \lambda^*.
\end{aligned}
$$

By Lemma 4.1, we know that $K^*_{\mu}(A) \leq \mathrm{D}^{h,\mathcal{H}}_{\phi}(\nu\|\mu)$. This indicates that

$$
\mathrm{D}^{h,\mathcal{H}}_{\phi}(\nu\|\mu) \geq \sup_{t \in \mathbb{R}} tA - K_{\mu}(t) \geq \sup_{t \geq 0} tA - K_{\mu}(t).
$$

*We hint that with more careful handling of the aforementioned step, one can obtain a bound for the absolute mean deviation (i.e. $|A|$) in the end. However, the current development suffices for our immediate objectives.*

Notice that when $t = 0$, this holds trivially. When $t > 0$, the above inequality is equivalent to

$$
A \leq \inf_{t > 0} \frac{1}{t}\left(K_{\mu}(t) + \mathrm{D}^{h,\mathcal{H}}_{\phi}(\nu\|\mu)\right).
$$

Hence, we have

$$
A \leq \inf_{t \geq 0} \frac{1}{t}\left(K_{\mu}(t) + \mathrm{D}^{h,\mathcal{H}}_{\phi}(\nu\|\mu)\right).
$$

Plugging the bound for $A$ into the inequality at the beginning of the proof, we obtain the first desired result

$$R_\nu(h) - R_\mu(h) \le \inf_{t \ge 0} \frac{1}{t}\left(K_\mu(t) + D_\phi^{h,\mathcal{H}}(\nu||\mu)\right) + \lambda^*.$$

For the second part, we apply Lemma B.4 here, then it is easy to see that $K_\mu(t) \le \frac{t^2}{2\phi''(1)}$. Therefore,

$$R_\nu(h) - R_\mu(h) \le \inf_t \frac{t}{2\phi''(1)} + \frac{D_\phi^{h,\mathcal{H}}(\nu||\mu)}{t} + \lambda^* = \sqrt{\frac{2D_\phi^{h,\mathcal{H}}(\nu||\mu)}{\phi''(1)}} + \lambda^*.$$

where the last equality is obtained by letting $t = \sqrt{2\phi''(1)D_\phi^{h,\mathcal{H}}(\nu||\mu)}$. This completes the proof. $\square$

### D.6  Proof of Corollary 4.1

**Corollary 4.1.** *Let $\ell \in [0,1]$, then for any $h \in \mathcal{H}$, we have $R_\nu(h) \le R_\mu(h) + \sqrt{2D_{\mathrm{KL}}^{h,\mathcal{H}}(\nu||\mu)} + \lambda^*$.*

*Proof.* In the case of KL, $\phi(x) = x\log x - x + 1$, and $1/\phi''(x) = x$. Hence, this corollary can be directly obtained from Eq. (4) in Theorem 4.1 by substituting $\phi''(1) = 1$. $\square$

### D.7  Proof of Lemma 4.2 and Generalization Bounds for $f$-DD

**Lemma 4.2.** *Let $\ell \in [0,1]$ and let $t_0$ be the optimal $t$ achieving the superum in $D_\phi^{h,\mathcal{H}}(\nu||\mu)$. Assume $\phi^*$ is $L$-Lipschitz, then for any given $h$, with probability at least $1 - \delta$, we have*

$$D_\phi^{h,\mathcal{H}}(\nu||\mu) \le D_\phi^{h,\mathcal{H}}(\hat\nu||\hat\mu) + 2\,|t_0|\,\hat{\mathfrak{R}}_{\mathcal{T}}(\mathcal{H}^\ell) + 2L\,|t_0|\,\hat{\mathfrak{R}}_{\mathcal{S}}(\mathcal{H}^\ell) + \mathcal{O}\left(\sqrt{\frac{\log(1/\delta)}{n}} + \sqrt{\frac{\log(1/\delta)}{m}}\right).$$

*Proof of Lemma 4.2.* For a fixed $h$,

$$D_\phi^{h,\mathcal{H}}(\nu||\mu) - D_\phi^{h,\mathcal{H}}(\hat\nu||\hat\mu)$$

$$= \sup_{h'}\sup_\alpha \mathbb{E}_\nu\left[t_0\ell(h,h')\right] + \mathbb{E}_\mu\left[\alpha - \phi^*(t_0\ell(h,h') + \alpha)\right] - \left(\sup_{h',t}\sup_\alpha \mathbb{E}_{\hat\nu}\left[t\ell(h,h')\right] + \mathbb{E}_{\hat\mu}\left[\alpha - \phi^*(t\ell(h,h') + \alpha)\right]\right)$$

$$\le \sup_{h'}\sup_\alpha \mathbb{E}_\nu\left[t_0\ell(h,h')\right] + \mathbb{E}_\mu\left[\alpha - \phi^*(t_0\ell(h,h') + \alpha)\right] - \left(\mathbb{E}_{\hat\nu}\left[t_0\ell(h,h')\right] + \mathbb{E}_{\hat\mu}\left[\alpha - \phi^*(t_0\ell(h,h') + \alpha)\right]\right)$$

$$\le \sup_{h'}\left|\mathbb{E}_\nu\left[t_0\ell(h,h')\right] - \mathbb{E}_{\hat\nu}\left[t_0\ell(h,h')\right]\right| + \sup_{h'}\sup_\alpha \left|\mathbb{E}_\mu\left[\phi^*(t_0\ell(h,h') + \alpha)\right] - \mathbb{E}_{\hat\mu}\left[\phi^*(t_0\ell(h,h') + \alpha)\right]\right|$$

$$\le 2\,|t_0|\,\hat{\mathfrak{R}}_{\mathcal{T}}(\mathcal{H}^\ell) + \mathcal{O}\left(\sqrt{\frac{\log(1/\delta)}{m}}\right) + 2\hat{\mathfrak{R}}_{\mathcal{S}}(\phi^* \circ \mathcal{H}^{t_0\ell}) + \mathcal{O}\left(\sqrt{\frac{\log(1/\delta)}{n}}\right)$$

$$\le 2\,|t_0|\,\hat{\mathfrak{R}}_{\mathcal{T}}(\mathcal{H}^\ell) + 2L\,|t_0|\,\hat{\mathfrak{R}}_{\mathcal{S}}(\mathcal{H}^\ell) + \mathcal{O}\left(\sqrt{\frac{\log(1/\delta)}{n}} + \sqrt{\frac{\log(1/\delta)}{m}}\right),$$

where the last two inequalities are by the scaling property of Rademacher complexity and Talagrand's lemma [70]. This completes the proof. $\square$

The generalization bound for $f$-DD is then given as follows, which clearly shows a slow convergence rate.

**Theorem D.1.** *Under the conditions in Lemma 4.2. Let $\phi$ be twice differentiable and $\phi''$ is monotone, for any $h \in \mathcal{H}$, with probability at least $1 - \delta$, we have*

$$R_\nu(h) \le R_{\hat\mu}(h) + \lambda^* + \mathcal{O}\left(\sqrt{D_\phi^{h,\mathcal{H}}(\hat\nu||\hat\mu)} + \sqrt{\hat{\mathfrak{R}}_{\mathcal{T}}(\mathcal{H}^\ell) + \hat{\mathfrak{R}}_{\mathcal{S}}(\mathcal{H}^\ell)}\right) + \mathcal{O}\left(\hat{\mathfrak{R}}_{\mathcal{S}}(\mathcal{H}^\ell)\right)$$

$$+ \mathcal{O}\left(\sqrt{\sqrt{\frac{\log(1/\delta)}{n}} + \sqrt{\frac{\log(1/\delta)}{m}}}\right) + \mathcal{O}\left(\sqrt{\frac{\log(1/\delta)}{n}}\right).$$

*Proof of Theorem D.1.* The concentration result for $R_\mu(h) - R_{\hat{\mu}}(h)$ is exactly the same as Eq. (18) in the proof of Theorem 4.1. Then putting everything together and by $\sqrt{\sum_{i=1}^n x_i} \leq \sum_{i=1}^n \sqrt{x_i}$ will complete the proof. □

## D.8 Proof of Theorem 5.1

**Theorem 5.1.** *For any $h \in \mathcal{H}_{r_1}$ and constants $C_1, C_2 \in (0, +\infty)$ satisfying $K_{h',\mu}(C_1) \leq C_1 C_2 \mathbb{E}_\mu [\ell(h, h')]$ for any $h' \in \mathcal{H}_r$, the following holds:*

$$R_\nu(h) \leq R_\mu(h) + \frac{1}{C_1} \mathrm{D}_\phi^{h, \mathcal{H}_r}(\nu||\mu) + C_2 R_\mu^r(h) + \lambda_r^*,$$

*where $\lambda_r^* = \min_{h^* \in \mathcal{H}_r} R_\mu(h^*) + R_\nu(h^*)$ and $R_\mu^r(h) = \sup_{h' \in \mathcal{H}_r} \mathbb{E}_\mu [\ell(h, h')]$.*

*Proof.* For a given $h \in \mathcal{H}_{r_1}$, let $h_r^* = \arg\min_{h^* \in \mathcal{H}_r} R_\mu(h^*) + R_\nu(h^*)$, then following similar steps in Lemma 3.1 and Theorem 4.1, we first have

$$R_\nu(h) - R_\mu(h) \leq \mathbb{E}_\nu [\ell(h, h_r^*)] - \mathbb{E}_\mu [\ell(h, h_r^*)] + \lambda_r^*.$$

Then, we can apply Lemma 4.1 for bounding $\mathbb{E}_\nu [\ell(h, h_r^*)] - \mathbb{E}_\mu [\ell(h, h_r^*)]$ here, which gives us

$$R_\nu(h) - R_\mu(h) \leq \inf_t \frac{1}{t} \left( K_\mu^r(t) + \mathrm{D}_\phi^{h, \mathcal{H}_r}(\nu||\mu) \right) + \lambda_r^*, \tag{24}$$

where $K_\mu^r(t) = \sup_{h' \in \mathcal{H}_r} K_{h',\mu}(t)$.

The condition for $C_1$ and $C_2$ to exist indicates that

$$K_\mu^r(C_1) = \sup_{h' \in \mathcal{H}_r} K_{h',\mu}(C_1) \leq C_1 C_2 R_\mu^r(h).$$

Replacing $t$ by $C_1$ and plugging the inequality above into Eq. (24) give us

$$R_\nu(h) - R_\mu(h) \leq \inf_{C_1, C_2} \frac{1}{C_1} \mathrm{D}_\phi^{h, \mathcal{H}_r}(\nu||\mu) + C_2 R_\mu^r(h) + \lambda_r^*,$$

which completes the proof. □

## D.9 Proof of Lemma 5.1

**Lemma 5.1.** *Let $\ell \in [0, 1]$, and let the constants $C_1 > 0$ and $C_2 \in (0, 1)$ satisfy the condition $\left(e^{C_1} - C_1 - 1\right)\left(1 - \min\{r_1 + r, 1\} + C_2^2 \min\{r_1 + r, 1\}\right) \leq C_1 C_2$. Then, for any $h \in \mathcal{H}_{r_1}$ and $h' \in \mathcal{H}_r$, we have*

$$\mathbb{E}_\nu [\ell(h, h')] \leq \inf_{C_1, C_2} \frac{\mathrm{D}_{\mathrm{KL}}^{h, \mathcal{H}_r}(\nu||\mu)}{C_1} + (1 + C_2)\mathbb{E}_\mu [\ell(h, h')].$$

*Proof.* Let $C' = 1 + C$ where $C \in (0, 1)$ and let $g(\cdot) = \ell(h(\cdot), h'(\cdot))$, then we aim to bound the weighted gap: $\mathbb{E}_\nu [g(X)] - C' \mathbb{E}_\mu [g(X)]$.

By definition,

$$\mathrm{D}_{\mathrm{KL}}^{h, \mathcal{H}_r}(\nu||\mu) = \sup_{g, t} t \mathbb{E}_\nu [g(X)] - \log \mathbb{E}_\mu \left[ e^{tg(X)} \right]$$

$$= \sup_{g, t} t \mathbb{E}_\nu [g(X)] - C' t \mathbb{E}_\mu [g(X)] - \log \mathbb{E}_\mu \left[ e^{t\left(g(X) - C' \mathbb{E}_\mu [g(X)]\right)} \right] \tag{25}$$

Then, recall that $\ell \leq 1$, and we use Lemma B.5[4] to obtain that

$$\log \mathbb{E}_\mu \left[ e^{t\left(g(X) - C' \mathbb{E}_\mu [g(X)]\right)} \right]$$

$$\leq t(1 - C') \mathbb{E}_\mu [g(X)] + B(t) t^2 \mathbb{E}_\mu \left[ \left(g(X) - C' \mathbb{E}_\mu [g(X)]\right)^2 \right]$$

$$= B(t) t^2 \mathbb{E}_\mu \left[ (g(X))^2 \right] + B(t) t^2 (C'^2 - 2C') \left(\mathbb{E}_\mu [g(X)]\right)^2 - t(C' - 1) \mathbb{E}_\mu [g(X)]$$

$$= B(t) t^2 \mathbb{E}_\mu \left[ (g(X))^2 \right] + B(t) t^2 (C^2 - 1) \left(\mathbb{E}_\mu [g(X)]\right)^2 - tC \mathbb{E}_\mu [g(X)], \tag{26}$$

---

[4]Note that, in this context, the random variable has a non-zero mean. Therefore, the first expectation term in Eq. (6) within the proof of Lemma B.5 should be retained.

where the function $B(\cdot)$ is the Bernstein function defined in Lemma B.5.

Since $0 \leq \mathbb{E}_\mu [g(X)] \leq \min\{r_1 + r, 1\}$ and $g(\cdot) \in [0, 1]$, we have

$$B(t)t\mathbb{E}_\mu \left[ (g(X))^2 \right] + B(t)t(C^2 - 1) \left( \mathbb{E}_\mu [g(X)] \right)^2 - C\mathbb{E}_\mu [g(X)]$$
$$\leq B(t)t\mathbb{E}_\mu [g(X)] + B(t)t(C^2 - 1) \min\{r_1 + r, 1\}\mathbb{E}_\mu [g(X)] - C\mathbb{E}_\mu [g(X)].$$

Our theme here is to obtain $\log \mathbb{E}_\mu \left[ e^{t\left(g(X) - C'\mathbb{E}_\mu[g(X)]\right)} \right] \leq 0$, so having the following inequality hold is sufficient

$$B(t)t\mathbb{E}_\mu [g(X)] + B(t)t(C^2 - 1) \min\{r_1 + r, 1\}\mathbb{E}_\mu [g(X)] - C\mathbb{E}_\mu [g(X)] \leq 0.$$

Therefore, $\log \mathbb{E}_\mu \left[ e^{t\left(g(X) - C'\mathbb{E}_\mu[g(X)]\right)} \right] \leq 0$ will hold if the following satisfies for $t$ and $C$,

$$B(t)t + B(t)t(C^2 - 1) \min\{r_1 + r, 1\} - C \leq 0.$$

Equivalently, substituting the expression of the Bernstein function $B(\cdot)$ gives us

$$\left( e^t - t - 1 \right) \left( 1 - \min\{r_1 + r, 1\} + C^2 \min\{r_1 + r, 1\} \right) - tC \leq 0, \tag{27}$$

Now let $t = C_1$ and $C = C_2$ satisfy Eq. (27), then for any $h, h'$, by re-arranging terms in Eq. (25), we have

$$\mathbb{E}_\nu [\ell(h, h')] - \mathbb{E}_\mu [\ell(h, h')] \leq \frac{1}{C_1} \left( D_{\mathrm{KL}}^{h, \mathcal{H}_r}(\nu || \mu) + \log \mathbb{E}_\mu \left[ e^{C_1(g(X) - (1+C_2)\mathbb{E}_\mu[g(X)])} \right] \right) + C_2 \mathbb{E}_\mu [\ell(h, h')]$$
$$\leq \frac{1}{C_1} D_{\mathrm{KL}}^{h, \mathcal{H}_r}(\nu || \mu) + C_2 \mathbb{E}_\mu [\ell(h, h')],$$

where the last inequality holds because $\log \mathbb{E}_\mu \left[ e^{C_1(g(X) - (1+C_2)\mathbb{E}_\mu[g(X)])} \right] \leq 0$ when Eq. (27) is satisfied.

This completes the proof. $\qquad \square$

### D.10  Proof of Theorem 5.2

**Theorem 5.2.** *Under the conditions in Lemma 5.1. For any $h \in \mathcal{H}_{r_1}$, with probability at least $1 - \delta$, we have*

$$R_\nu(h) \leq R_{\hat{\mu}}(h) + \frac{D_{\mathrm{KL}}^{h, \mathcal{H}_r}(\hat{\nu} || \hat{\mu})}{C_1} + C_2 R_\mu^r(h) + \mathcal{O}\left( \hat{\mathfrak{R}}_\mathcal{T} \left( \mathcal{H}_r^\ell \right) + \hat{\mathfrak{R}}_\mathcal{S} \left( \mathcal{H}_{\max\{r, r_1\}}^\ell \right) \right)$$
$$+ \mathcal{O}\left( \frac{\log(1/\delta)}{n} + \frac{\log(1/\delta)}{m} \right) + \mathcal{O}\left( \sqrt{\frac{(r_1 + r)\log(1/\delta)}{n}} + \sqrt{\frac{r \log(1/\delta)}{m}} \right) + \lambda_r^*.$$

*Proof.* We first apply both Lemma B.6 and Lemma B.3 for bounding $R_\mu(h)$ with local Rademacher Complexity $\hat{\mathfrak{R}}_\mathcal{S}(\mathcal{H}_{r_1}) = \mathbb{E}_{\varepsilon_{1:n}} \left[ \sup_{h \in \mathcal{H}_{r_1}} \frac{1}{n} \sum_{i=1}^n \varepsilon_i \ell(h, f_\mu) \right]$,

$$\sup_h R_\mu(h) - R_{\hat{\mu}}(h) \leq 2\mathbb{E}_{\mathcal{S} \sim \mu^{\otimes n}} \left[ \sup_h R_\mu(h) - R_{\hat{\mu}}(h) \right] + \mathcal{O}\left( \sqrt{\frac{r_1 \log(1/\delta)}{n}} + \frac{\log(1/\delta)}{n} \right)$$
$$\leq 2\hat{\mathfrak{R}}_\mathcal{S}(\mathcal{H}_{r_1}) + \mathcal{O}\left( \sqrt{\frac{r_1 \log(1/\delta)}{n}} + \frac{\log(1/\delta)}{n} \right), \tag{28}$$

where we use the fact that $\mathrm{Var}_\mu (\ell(h, f_\mu)) \leq R_\mu(h) \leq r_1$ for $\ell \in [0, 1]$.

Similar to Lemma D.1, let $t = t_0$ achieve the supreme in $D_{\mathrm{KL}}^{h, \mathcal{H}_r}(\nu || \mu)$. Let $\beta \leq \min\left\{ \mathbb{E}_\nu \left[ e^{\ell(h, h')} \right], \mathbb{E}_{\hat{\nu}} \left[ e^{\ell(h, h')} \right] \right\}$ for any $\hat{\nu}$ and $h, h' \in \mathcal{H}$ (e.g., $\beta = 1$), then the concentration

result of localized KL-DD is derived below,

$$\mathrm{D}_{\mathrm{KL}}^{h,\mathcal{H}_r}(\nu||\mu) - \mathrm{D}_{\mathrm{KL}}^{h,\mathcal{H}_r}(\hat{\nu}||\hat{\mu})$$

$$\leq \sup_{h'} |\mathbb{E}_\nu\left[t_0\ell(h,h')\right] - \mathbb{E}_{\hat{\nu}}\left[t_0\ell(h,h')\right]| + \sup_{h'}\left|\log\mathbb{E}_\mu\left[e^{t_0\ell(h,h')}\right] - \log\mathbb{E}_{\hat{\mu}}\left[e^{t_0\ell(h,h')}\right]\right|$$

$$\leq \sup_{h'} |\mathbb{E}_\nu\left[t_0\ell(h,h')\right] - \mathbb{E}_{\hat{\nu}}\left[t_0\ell(h,h')\right]| + \sup_{h'}\frac{1}{\beta}\left|\mathbb{E}_\mu\left[e^{t_0\ell(h,h')}\right] - \mathbb{E}_{\hat{\mu}}\left[e^{t_0\ell(h,h')}\right]\right| \qquad (29)$$

$$\leq 2\,|t_0|\,\hat{\mathfrak{R}}_{\mathcal{T}}(\mathcal{H}_r^\ell) + \mathcal{O}\left(\sqrt{\frac{r\log(1/\delta)}{m}} + \frac{\log(1/\delta)}{m}\right) + 2\frac{e}{\beta}\,|t_0|\,\hat{\mathfrak{R}}_{\mathcal{S}}(\mathcal{H}_r^\ell) + \mathcal{O}\left(\sqrt{\frac{r\log(1/\delta)}{n}} + \frac{\log(1/\delta)}{n}\right),$$
$$\tag{30}$$

where Eq. (29) follows from the similar developments in Eq. (16).

Putting Eq. (30), Eq. (28), Lemma 5.1 and Theorem 5.1 all together, we have

$$R_\nu(h) \leq R_{\hat{\mu}}(h) + \inf_{C_1,C_2}\frac{\mathrm{D}_{\mathrm{KL}}^{h,\mathcal{H}_r}(\hat{\nu}||\hat{\mu})}{C_1} + C_2 R_\mu^r(h) + \mathcal{O}\left(\hat{\mathfrak{R}}_{\mathcal{S}}\left(\mathcal{H}_{r_1}^\ell\right) + \hat{\mathfrak{R}}_{\mathcal{S}}\left(\mathcal{H}_r^\ell\right) + \hat{\mathfrak{R}}_{\mathcal{T}}\left(\mathcal{H}_r^\ell\right)\right)$$

$$+ \mathcal{O}\left(\frac{\log(1/\delta)}{n} + \frac{\log(1/\delta)}{m}\right) + \mathcal{O}\left(\sqrt{\frac{r_1\log(1/\delta)}{n}} + \sqrt{\frac{r\log(1/\delta)}{n}} + \sqrt{\frac{r\log(1/\delta)}{m}}\right) + \lambda_r^*.$$

Finally, using $\sqrt{a} + \sqrt{b} \leq 2\sqrt{a+b}$ and $\mathcal{O}\left(\hat{\mathfrak{R}}_{\mathcal{S}}\left(\mathcal{H}_{r_1}^\ell\right) + \hat{\mathfrak{R}}_{\mathcal{S}}\left(\mathcal{H}_r^\ell\right)\right) = \mathcal{O}\left(\hat{\mathfrak{R}}_{\mathcal{S}}\left(\mathcal{H}_{\max\{r_1,r\}}^\ell\right)\right)$ will conclude the proof. □

### D.11    Generalization Bounds based on $\chi^2$-DD

**Theorem D.2.** *Let* $\ell(\cdot,\cdot) \in [0,1]$. *For any* $h \in \mathcal{H}_{r_1}$, $C_1 > 0$ *and* $C_2 \in (0,1)$ *satisfying* $\frac{C_1\mathrm{Var}_\mu\left(\ell(h,h')\right)}{4} \leq C_2\mathbb{E}_\mu\left[\ell(h,h')\right]$ *for any* $h' \in \mathcal{H}_r$, *with probability at least* $1-\delta$, *we have*

$$R_\nu(h) \leq R_{\hat{\mu}}(h) + \frac{\mathrm{D}_{\chi^2}^{h,\mathcal{H}_r}(\hat{\nu}||\hat{\mu})}{C_1} + C_2 R_\mu^r(h) + \mathcal{O}\left(\hat{\mathfrak{R}}_{\mathcal{T}}\left(\mathcal{H}_r^\ell\right) + \hat{\mathfrak{R}}_{\mathcal{S}}\left(\mathcal{H}_{\max\{r,r_1\}}^\ell\right)\right) + \lambda_r^*$$

$$+ \mathcal{O}\left(\frac{\log(1/\delta)}{n} + \frac{\log(1/\delta)}{m}\right) + \mathcal{O}\left(\sqrt{\frac{(r_1+r)\log(1/\delta)}{n}} + \sqrt{\frac{r\log(1/\delta)}{m}}\right).$$

*Proof.* We first give a similar result to Lemma 5.1 based on $\chi^2$-DD.

Again, let $g(\cdot) = \ell(h,h')$. By Eq. (8), we know that our localized $\chi^2$-DD is

$$\mathrm{D}_{\chi^2}^{h,\mathcal{H}_r}(\nu||\mu) = \sup_{h',t} t\left(\mathbb{E}_\nu\left[g(X)\right] - \mathbb{E}_\mu\left[g(X)\right]\right) - \frac{t^2\mathrm{Var}_\mu\left(g(X)\right)}{4}.$$

Let $C' = 1 + C$ where $C \in (0,1)$. Then, for any $h, h'$,

$$\mathbb{E}_\nu\left[g(X)\right] - C'\mathbb{E}_\mu\left[g(X)\right] \leq \inf_{t\geq 0}\frac{1}{t}\mathrm{D}_{\chi^2}^{h,\mathcal{H}_r}(\nu||\mu) + \frac{t\mathrm{Var}_\mu\left(g(X)\right)}{4} - C\mathbb{E}_\mu\left[g(X)\right].$$

When $t$ and $C$ satisfy $\frac{t\mathrm{Var}_\mu(g(X))}{4} - C\mathbb{E}_\mu\left[g(X)\right] \leq 0$, we have $\mathbb{E}_\nu\left[g(X)\right] - \mathbb{E}_\mu\left[g(X)\right] \leq \inf_{t,C}\frac{1}{t}\mathrm{D}_{\chi^2}^{h,\mathcal{H}_r}(\nu||\mu) + C\mathbb{E}_\mu\left[g(X)\right]$.

For the concentration result,

$$\mathrm{D}_{\chi^2}^{h,\mathcal{H}_r}(\nu||\mu) - \mathrm{D}_{\chi^2}^{h,\mathcal{H}_r}(\hat{\nu}||\hat{\mu})$$

$$\leq \sup_{h',t} t\left(\mathbb{E}_\nu\left[g(X)\right] - \mathbb{E}_\mu\left[g(X)\right]\right) - \frac{t^2\mathrm{Var}_\mu\left(g(X)\right)}{4} - t\left(\mathbb{E}_{\hat{\nu}}\left[g(X)\right] - \mathbb{E}_{\hat{\mu}}\left[g(X)\right]\right) + \frac{t^2\mathrm{Var}_{\hat{\mu}}\left(g(X)\right)}{4}$$

$$= \sup_{h',t} t\left(\mathbb{E}_\nu\left[g(X)\right] - \mathbb{E}_{\hat{\nu}}\left[g(X)\right]\right) + t\left(\mathbb{E}_{\hat{\mu}}\left[g(X)\right] - \mathbb{E}_\mu\left[g(X)\right]\right) + \frac{t^2}{4}\left(\mathrm{Var}_{\hat{\mu}}\left(g(X)\right) - \mathrm{Var}_\mu\left(g(X)\right)\right)$$

$$\leq 2\left|t_0\right|\hat{\mathfrak{R}}_\mathcal{S}(\mathcal{H}_r) + 2\left|t_0\right|\hat{\mathfrak{R}}_\mathcal{T}(\mathcal{H}_r) + \mathcal{O}\left(\sqrt{\frac{r_1\log(1/\delta)}{n}} + \frac{\log(1/\delta)}{n} + \sqrt{\frac{r_1\log(1/\delta)}{m}} + \frac{\log(1/\delta)}{m}\right)$$

$$+ \sup_{h'} \frac{t_0^2}{4}\left(\mathrm{Var}_{\hat{\mu}}\left(g(X)\right) - \mathrm{Var}_\mu\left(g(X)\right)\right).$$

Notice that

$$\sup_{h'} \frac{t_0^2}{4}\left(\mathrm{Var}_{\hat{\mu}}\left(g(X)\right) - \mathrm{Var}_\mu\left(g(X)\right)\right)$$

$$= \sup_{h'} \frac{t_0^2}{4}\left(\mathbb{E}_{\hat{\mu}}\left[g^2(X)\right] - \mathbb{E}_\mu\left[g^2(X)\right] + \left(\mathbb{E}_\mu\left[g(X)\right]\right)^2 - \left(\mathbb{E}_{\hat{\mu}}\left[g(X)\right]\right)^2\right)$$

$$= \sup_{h'} \frac{t_0^2}{4}\left(\mathbb{E}_{\hat{\mu}}\left[g^2(X)\right] - \mathbb{E}_\mu\left[g^2(X)\right] + \left(\mathbb{E}_\mu\left[g(X)\right] - \mathbb{E}_{\hat{\mu}}\left[g(X)\right]\right)\left(\mathbb{E}_\mu\left[g(X)\right] + \mathbb{E}_{\hat{\mu}}\left[g(X)\right]\right)\right)$$

$$\leq \sup_{h'} \frac{t_0^2}{4}\mathbb{E}_{\hat{\mu}}\left[g^2(X)\right] - \mathbb{E}_\mu\left[g^2(X)\right] + \sup_{h'} \frac{t_0^2}{2}\left(\mathbb{E}_\mu\left[g(X)\right] - \mathbb{E}_{\hat{\mu}}\left[g(X)\right]\right)$$

$$\leq t_0^2\hat{\mathfrak{R}}_\mathcal{S}(\mathcal{H}_r) + \mathcal{O}\left(\sqrt{\frac{r\log(1/\delta)}{n}} + \frac{\log(1/\delta)}{n}\right),$$

where we use the fact that $g^2$ is 2-Lipschitz in $g \in [0,1]$.

Notice that as mentioned in Remark 4.2, $t_0 = \frac{2\left(\mathbb{E}_\nu\left[\ell(h,h'^*)\right] - \mathbb{E}_\mu\left[\ell(h,h'^*)\right]\right)}{\mathrm{Var}_\mu(\ell(h,h'^*))}$. Putting everything together and following the last several steps as in Theorem 5.2 will complete the proof. □

### D.12 Threshold Learning Example

Consider a popular threshold learning example.

**Example 1** (Threshold Learning). Let $\mathcal{X} = \mathbb{R}$. A threshold function, denoted as $h_c$, is defined s.t. $h_c(x) = 0$ if $x < c$ and $h_c(x) = 1$ if $x \geq c$. Let the source domain $\mu$ be an uniform distribution on $[0,1]$ and let the target domain $\nu$ be another uniform distribution on $[0,2]$. Let $\mathcal{H} = \{h_c | c \in [0, \frac{1}{2}]\}$ and assume the ground-truth hypothesis is $h_{\frac{1}{2}}^*$ for the both the source and target domains. Let $\ell$ be the zero-one loss.

Recall that $\mathrm{D}_{\mathrm{KL}}^{h,\mathcal{H}}(\nu||\mu) = \sup_{h',t} t\mathbb{E}_\nu[\ell(h,h')] - \log\mathbb{E}_\mu e^{t\ell(h,h')}$. Hence,

$$\mathrm{D}_{\mathrm{KL}}^{h_{\frac{1}{2}},\mathcal{H}}(\nu||\mu) = \sup_t \int_0^{\frac{1}{2}} \frac{t}{2}\cdot 1 dx - \log\left(\int_0^{\frac{1}{2}} e^t dx + \int_{\frac{1}{2}}^1 e^0 dx\right) = \sup_t \frac{t}{4} - \log\left(\frac{1}{2} + \frac{e^t}{2}\right) = 0.131.$$

Now let $r = 1/4$, so $\mathcal{H}_r = \{h_c | c \in [\frac{1}{4}, \frac{1}{2}]\}$, by a similar calculation we have

$$\mathrm{D}_{\mathrm{KL}}^{h_{\frac{1}{2}},\mathcal{H}_{\frac{1}{4}}}(\nu||\mu) = \sup_t \frac{t}{8} - \log(\frac{3}{4} + \frac{e^t}{4}) = 0.048 \text{ and } R_\mu^r = \frac{1}{4}.$$

Based on Remark 5.3, we can set $C_2 = 0.1$ and $C_1 = 3.74$. Hence, we have

$$0.27 \cdot \mathrm{D}_{\mathrm{KL}}^{h_{\frac{1}{2}},\mathcal{H}_{\frac{1}{4}}}(\nu||\mu) + 0.1 \cdot R_\mu^r = 0.038 \leq \sqrt{\mathrm{D}_{\mathrm{KL}}^{h_{\frac{1}{2}},\mathcal{H}}(\nu||\mu)} = 0.36.$$

Since $h_{\frac{1}{2}}^* \in \mathcal{H}_r \subset \mathcal{H}$, we have $\lambda^* = \lambda_r^* = 0$ in this case. This example justifies that the localization technique can significantly tighten the bound. Note that $C_2 = 0.1$ and $C_1 = 3.74$ are not the optimal choice so localized bound can be even tighter.

In addition, as a more extreme case, let $r \to 0$ so $\mathcal{H}_r = \{h_{\frac{1}{2}}\}$. Then by a similar calculation, we have $\mathrm{D}_{\mathrm{KL}}^{h_{\frac{1}{2}}, \mathcal{H}_0}(\nu||\mu) = R_\mu^r = 0$, while $\mathrm{D}_{\mathrm{KL}}^{h_{\frac{1}{2}}, \mathcal{H}}(\nu||\mu)$ remains unchanged. Notice that $R_\nu(h_{\frac{1}{2}}) - R_\mu(h_{\frac{1}{2}}) = 0$ so localized $f$-DD gives a tightest target error bound in this case.

### D.13 Proof of Proposition 1

**Proposition 1.** *Let* $d_{\hat{\mu}, \hat{\nu}}(h, h') = \mathbb{E}_{\hat{\nu}}\left[\hat{\ell}(h, h')\right] - I_{\phi, \hat{\mu}}^h(\hat{\ell} \circ h')$. *Assume* $\mathcal{H}$ *is sufficiently large s.t.* $\hat{\ell} : \mathcal{X} \to \mathbb{R}$, *we have* $\max_{h'} \tilde{d}_{\hat{\mu}, \hat{\nu}}(h, h') = \max_{h'} d_{\hat{\mu}, \hat{\nu}}(h, h') = \mathrm{D}_\phi^{h, \mathcal{H}'}(\hat{\nu}||\hat{\mu}) \le \mathrm{D}_\phi(\hat{\nu}||\hat{\mu})$.

*Proof.* Since $\hat{\ell} : \mathcal{X} \to \mathbb{R}$, $t\ell : \mathcal{X} \to \mathbb{R}$ and $t\ell + \alpha : \mathcal{X} \to \mathbb{R}$, then it is straightforward to have $\max_{h'} \tilde{d}_{\hat{\mu}, \hat{\nu}}(h, h') = \max_{h'} d_{\hat{\mu}, \hat{\nu}}(h, h') = \mathrm{D}_\phi^{h, \mathcal{H}'}(\hat{\mu}||\hat{\nu})$. Additionally, as $t\ell \circ \mathcal{H}' \subseteq \mathcal{G}$, they are upper bounded by $\mathrm{D}_\phi(\hat{\mu}||\hat{\nu})$. $\qquad\square$

## E   More Experiment Details and Additional Results

We adopt the experimental setup from [1] and build upon their publicly available code, accessible at https://github.com/nv-tlabs/fDAL/tree/main. Additionally, we lever-age some settings from the implementations of [9] and [49], which can be found at https://github.com/thuml/MDD/tree/master and https://github.com/thuml/CDAN, respectively. Our code is available at https://github.com/ZiqiaoWangGeothe/f-DD.

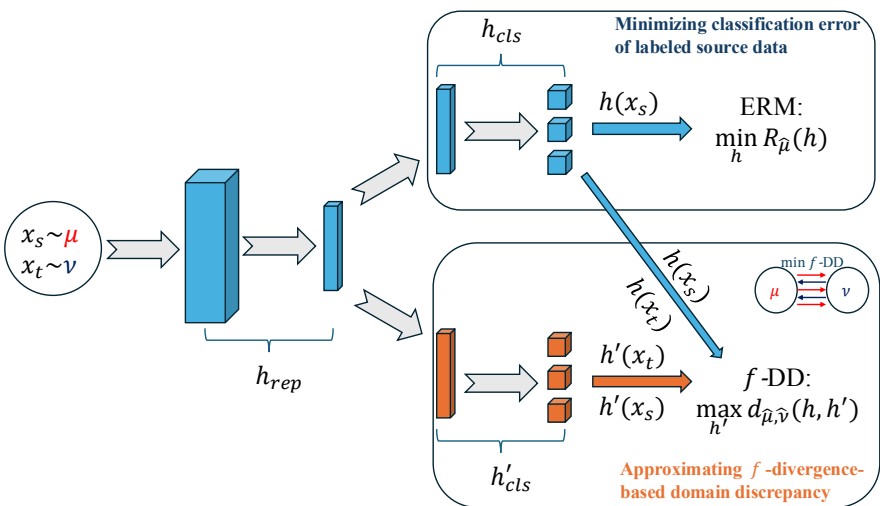

Figure 2: Illustration of the adversarial training framework for $f$-DD-based UDA. The framework includes the representation network ($h_{\mathrm{rep}}$), the main classifier ($h_{\mathrm{cls}}$), and the auxiliary classification network ($h'_{\mathrm{cls}}$). It jointly minimizes the empirical risk on the source domain and the approximated $f$-DD between the source and target domains.

Specifically, for experiments on Office-31 and Office-Home, we utilize a pretrained ResNet-50 backbone on ImageNet [74]. Our $f$-DD is trained for 40 epochs using SGD with Nesterov Momentum, setting the momentum to 0.9, the learning rate to 0.004, and the batch size to 32. Hyperparameter settings and training protocols closely align with [9] and [49]. Particularly, on Office-31, we vary the trade-off parameter $\eta$ for our KL-DD within $[3, 4.5, 5.75]$, and for our $\chi^2$-DD within $[1, 1.75, 2]$. For Jeffreys-DD, we choose $\eta\gamma_1, \eta\gamma_2$ from $[0, 1, 1.2, 3, 3.75, 4.5, 5, 5.75]$. On Office-Home, $\eta$ for KL-DD is chosen from $[3, 3.75, 4.5]$, $\eta$ for $\chi^2$-DD from $[3, 3.75]$, and $\eta\gamma_1, \eta\gamma_2$ for Jeffreys-DD from $[0, 1, 1.2, 3, 3.75, 4.5]$. For the Digits datasets, we train our $f$-DD for 30 epochs with SGD and

Nesterov Momentum, setting the momentum to 0.9 and the learning rate to 0.01. The batch size is set to 128 for $\mathbf{M} \rightarrow \mathbf{U}$ and 64 for $\mathbf{U} \rightarrow \mathbf{M}$. Other hyperparameters closely follow those used by [1] and [49]. The trade-off parameter $\eta$ for both KL-DD and $\chi^2$-DD is selected from $[0.75, 1]$, and $\eta\gamma_1, \eta\gamma_2$ for Jeffreys-DD from $[0, 0.01, 0.5, 0.55]$. All experiments are conducted on NVIDIA V100 (32GB) GPUs.

**Comparison with $f$-DAL + Implicit Alignment**    [1] also investigate the empirical performance of combining $f$-DAL with a sampling-based implicit alignment technique from [51], enhancing their original $f$-DAL. However, our findings reveal that our $f$-DD consistently outperforms $f$-DAL with implicit alignment, as detailed in Table 6. This observation suggests that the performance gain can be achieved by simply adopting a more tightly defined variational representation. Furthermore, in Table 6, the entry labeled $f$-DD (Best) corresponds to the average of the maximum accuracy across KL-DD, $\chi^2$-DD, and Jefferys-DD for each subtask. Notably, this aggregated result shows slight improvement on Office-31 when compared to the individual performance of Jefferys-DD.

Table 6: Comparison between $f$-DD and $f$-DAL

| Method | Office-31 | Office-Home |
|---|---|---|
| $f$-DAL | 89.5 | 68.5 |
| $f$-DAL+Imp. Align. | 89.2 | 70.0 |
| Jefferys-DD | 90.1 | **70.2** |
| $f$-DD (Best) | **90.3** | **70.2** |

**Ablation Study**    We adjust the trade-off hyper-parameter $\eta$ in KL-DD and present the outcomes for Office-31 and Office-Home. As depicted in Table 7 and Table 8, we can see that the performance exhibits relatively low sensitivity to changes in $\eta$. Hence, the tuning process need not be overly meticulous.

Table 7: Ablation Study on Office-31 for KL-DD

| $\eta$ | 3.75 | 4.5 | 5.5 | 5.75 |
|---|---|---|---|---|
| $\mathbf{A} \rightarrow \mathbf{D}$ | 95.5±0.7 | 95.4±0.4 | 95.4±0.7 | 95.9±0.6 |
| $\mathbf{W} \rightarrow \mathbf{A}$ | 74.5±0.1 | 74.6±0.7 | 74.6±0.4 | 74.5±0.5 |

Table 8: Ablation Study on Office-Home for KL-DD

| $\eta$ | 3 | 3.75 | 4.5 |
|---|---|---|---|
| $\mathbf{Ar} \rightarrow \mathbf{Cl}$ | 55.3±0.4 | 54.9±0.2 | 55.3±0.1 |
| $\mathbf{Pr} \rightarrow \mathbf{Rw}$ | 80.7±0.1 | 80.8±0.2 | 80.9±0.1 |

**Additional Results for Absolute Divergence**    In addition to Figure 1, we present the performance of the absolute KL discrepancy in Figure 3(a-b). The consistent observations persist, indicating that the absolute version of the discrepancy tends to overestimate the $f$-divergence, leading to a breakdown in the training process.

Table 9: Comparison between $\chi^2$-DD and Opt$\chi^2$-DD

| Method | $\mathbf{A} \rightarrow \mathbf{D}$ | $\mathbf{Ar} \rightarrow \mathbf{Cl}$ |
|---|---|---|
| $\chi^2$-DD | 95.0±0.4 | 55.2±0.3 |
| Opt$\chi^2$-DD | 93.1±0.3 | 53.9±0.3 |

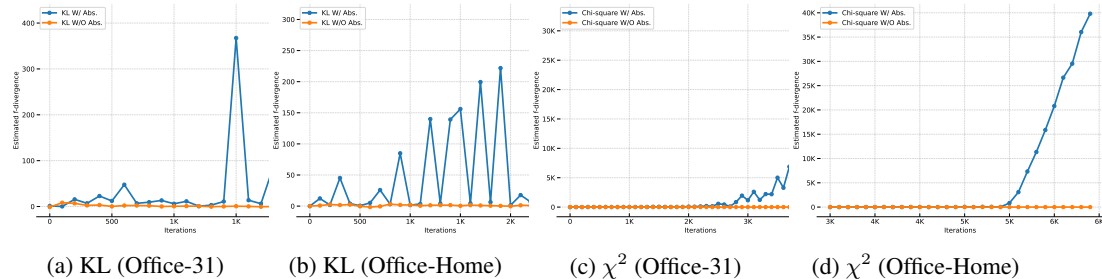

(a) KL (Office-31)  (b) KL (Office-Home)  (c) $\chi^2$ (Office-31)  (d) $\chi^2$ (Office-Home)

Figure 3: Comparison between $\mathrm{D}_\phi^{h,\mathcal{H}}$ and $\widetilde{\mathrm{D}}_\phi^{h,\mathcal{H}}$. The $y$-axis is the estimated corresponding $f$-divergence and the $x$-axis is the number of iterations.

**Details and Additional Results for Optimal $f$-DD**  As outlined in Section 6, instead of invoking a stochastic gradient-based optimizer (e.g., SGD) to update $t$, we utilize a quadratic approximation for the optimal $t$ as presented in [23]. The approximately optimal KL-DD (OptKL-DD) is expressed as follows:

$$\mathrm{D}_{\mathrm{KL}}^{h,\mathcal{H}}(\nu||\mu) = \sup_{h'} \left(1 + \Delta t^*\right) \mathbb{E}_\nu \left[\ell(h,h')\right] - \log \mathbb{E}_\mu \left[e^{(1+\Delta t^*)\ell(h,h')}\right].$$

Here, a Gibbs measure is defined as $d\mu' \triangleq \frac{e^{\ell(h,h')}d\mu}{\mathbb{E}_\mu\left[e^{\ell(h,h')}\right]}$, and $\Delta t^*$ is determined by the formula

$$\Delta t^* = \frac{\mathbb{E}_\nu \left[\ell(h,h')\right] - \mathbb{E}_{\mu'} \left[\ell(h,h')\right]}{\mathrm{Var}_{\mu'}(\ell(h,h'))}.$$

This approximation is obtained by [23] through a Taylor expansion around $t = 1$, with additional details provided in their Appendix B. In our implementation of OptKL-DD, similar to KL-DD, we use $\hat{\ell}$ to replace $\ell$.

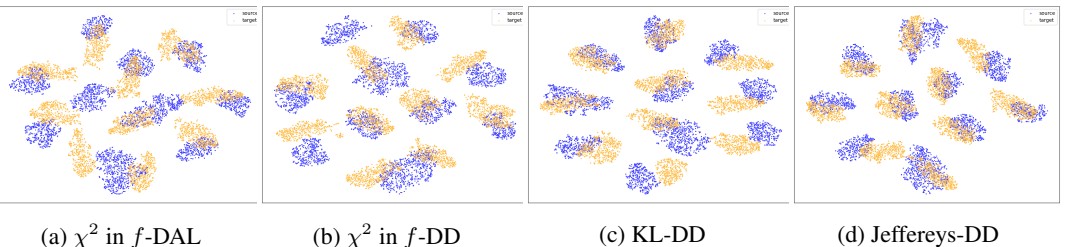

(a) $\chi^2$ in $f$-DAL  (b) $\chi^2$ in $f$-DD  (c) KL-DD  (d) Jeffereys-DD

Figure 4: Visualization results of representations obtained by using t-SNE. The source domain (blue points) is **U** and the target domain (orange points) is **M**.

For the optimal $\chi^2$-DD (Opt$\chi^2$-DD), we have its analytic form, as shown in Appendix C:

$$\mathrm{D}_{\chi^2}^{h,\mathcal{H}}(\nu||\mu) = \sup_{h'} \frac{\left(\mathbb{E}_\nu \left[\ell(h,h')\right] - \mathbb{E}_\mu \left[\ell(h,h')\right]\right)^2}{\mathrm{Var}_\mu \left(\ell(h,h')\right)}.$$

By replacing $\ell$ by $\hat{\ell}$, we utilize the above as the training objective in our implementation. Table 9 illustrates that the optimal form does not improve performance in two sub-tasks, namely **A**→**D** and **Ar**→**Cl**, and in fact, may even degrade performance. Consequently, when the surrogate loss itself is unbounded, optimizing over $t$ may not be necessary, at least for $\chi^2$-DD and KL-DD.

**Visualization Results**  To visualize model representations (output of $h_{\mathrm{rep}}$) trained with $f$-DAL, $\chi^2$-DD, KL-DD, and Jeffereys-DD, we leverage t-SNE [52]. In Figure 4, we present visualization results using USPS (**U**) as the source domain and MNIST (**M**) as the target domain. Notably, while the original $f$-DAL already provides satisfactory results, our $f$-DD achieves further improvements in representation alignment.

**More Advance Network Architecture** While our empirical study primarily aims to compare with [1] and improve their algorithms, in this section, we also evaluate our $f$-DD on more complicated models, which can serve as a valuable reference for practitioners.

In particular, we update our original backbone, ResNet-50, with pretrained transformer-based models. Specifically, we use the pretrained Vision transformer (ViT) base model (vit-base-patch16-224) [75] and pretrained Swin-based transformer (swin-base-patch4-window7-224) [76], while keeping all other settings in our algorithms unchanged. The results on Office-31 are presented in Table 10.

Table 10: Accuracy (%) on the Office-31 benchmark.

| Method | A → W | D → W | W → D | A → D | D → A | W → A | Avg |
|---|---|---|---|---|---|---|---|
| ViT-based [75] | 91.2 | 99.2 | 100.0 | 90.4 | 81.1 | 80.6 | 91.1 |
| SSRT-ViT [77] | 97.7 | 99.2 | 100.0 | 98.6 | 83.5 | 82.2 | 93.5 |
| PMTrans-ViT [78] | 99.1 | 99.6 | 100.0 | 99.4 | 85.7 | 86.3 | **95.0** |
| Ours-ViT | 98.0 ± 0.2 | 99.2 ± 0.0 | 100 ± 0.0 | 98.6 ± 0.2 | 83.5 ± 0.5 | 83.9 ± 0.4 | 93.9 |
| Swin-based [76] | 97.0 | 99.2 | 100.0 | 95.8 | 82.4 | 81.8 | 92.7 |
| PMTrans-Swin [78] | 99.5 | 99.4 | 100.0 | 99.8 | 86.7 | 86.5 | **95.3** |
| Ours-Swim | 98.7 ± 0.2 | 99.2 ± 0.0 | 100 ± 0.0 | 99.2 ± 0.2 | 86.1 ± 0.1 | 85.6 ± 0.3 | 94.8 |

Results on models other than ours are taken directly from [78, Table 2]. Here ViT-based and Swin-based refer to source-only training using ViT and Swin transformers, respectively, and "Ours" particularly refers to our weighted-Jeffereys discrepancy. While our method does not outperform the current SOTA, PMTrans, (especially for ViT), but close to it, this should demonstrate the practical utility of the proposed approach. Notably, our approach not only significantly improves the respective backbones but also outperforms the strongest baseline, SSRT-ViT [77], as compared in [78].

The fact the our approach does not outperform PMTrans should come at no surprise. In particular, PMTrans and all other compared models all include while our approach not only neglected these ingredients but also simply use hyperparameter based on our earlier ResNet-50 settings without further tuning.

One notable difference between our method and other SOTA approaches is the absence of any pseudo-labeling strategy in our framework. Specifically, our theoretical framework is constructed under the assumption of zero knowledge of target labels. In a strategy that successfully exploits pseudo-labelling, usually explicit or implicit prior knowledge are available (e.g., for the purpose of selecting hyperparameters relating to pseudo labeling). For example, such knowledge may present itself as a labeled validation set in the target domain. Such knowledge, if available, may change our problem setup and require a refinement of our theory. We anticipate that combining our method with advanced pseudo-labeling strategies, data augmentation techniques like Mixup (which itself inherently relies on pseudo-labels), and label smoothing will improve our results, possibly approaching the current SOTA performance.

