# OpenReview forum: "On $f$-Divergence Principled Domain Adaptation: An Improved Framework"
_NeurIPS.cc/2024/Conference — NeurIPS 2024 poster_

### Official Review · Reviewer_vWvG · 2024-07-01

**Soundness:** 3
**Presentation:** 3
**Contribution:** 3
**Rating:** 6
**Confidence:** 3

**Summary:**

This study addresses the gap in the theory and algorithms of unsupervised domain adaptation based on f-divergence proposed by Acuna et al. 2021. Specifically, while the theory uses absolute values, the algorithms do not, and this issue is resolved by introducing a single scaling factor. The newly proposed f-DD generalization bound is derived based on Rademacher complexity, and tighter bounds are obtained using the localization technique.

As a specific domain adaptation algorithm, an adversarial type algorithm is proposed, yielding favorable results in benchmarks.

**Strengths:**

This study bridges the gap between theory and practice in existing UDA methods based on f-divergence, advancing the foundational research in domain adaptation. Furthermore, the derivation of sharper bounds using the recently introduced localization technique for DA is highly commendable as a contribution to the theoretical framework of DA.
The authors validate their theoretical contributions with empirical results, showing superior performance on popular benchmarks.

**Weaknesses:**

While the empirical validation is strong, it is limited to specific benchmarks. Broader validation across diverse datasets and tasks would strengthen the findings. It is nice to present some insight into what kind of dataset the proposed f-DD works well (and why), and also into what kind of dataset it does not work well (and why).

**Questions:**

Please provide the source of the technique that resolves the absolute value with a scaling parameter.
Additionally, clarify whether the claim in this paper—that there is no need to adjust the scaling parameter—holds generally.

**Limitations:**

Not applicable.

---

> ### Author Rebuttal · Authors · 2024-08-06
>
> We thank you sincerely for the valuable feedback on our paper. Our responses follow.
>
>
> >- While the empirical validation is strong, it is limited to specific benchmarks. Broader validation across diverse datasets and tasks would strengthen the findings. It is nice to present some insight into what kind of dataset the proposed f-DD works well (and why), and also into what kind of dataset it does not work well (and why).
>
> **Response.** Thank you for your suggestion. We have included additional experimental results on the VisDA-2017 dataset in the updated PDF (please see our general response).
>
> Regarding when $f$-DD performs well or poorly: we believe $f$-DD tends to work effectively when the marginal distributions $P_X$ differ, but the conditional distribution $P_{Y|X}$ is similar or close across the source and target domains (e.g., in cases of covariate shift). This is because the calculation of prediction disagreement between domains relies on the marginal distributions. Conversely, if there is a significant difference in the conditional distributions $P_{Y|X}$ between the two domains, $f$-DD's effectiveness may be limited. However, such issue may exist in most domain discrepancy guided UDA algorithms.
> In such cases, utilizing additional knowledge about target domain labels, potentially through advanced pseudo-labeling techniques, might improve performance. We note that our theoretical framework assumes zero knowledge of target labels, so having such knowledge could require adjustments to our problem setup and refinement of our theory.
>
>
> >- Please provide the source of the technique that resolves the absolute value with a scaling parameter.
>
> **Response.** We believe that incorporating an absolute value function into the variational representation of $f$-divergence, as done by Acuna et al. (2021), is an unusual approach for upper bounding some objective quantities. This method changes the definition of $f$-divergence, as discussed at the beginning of our Section 4. In contrast, introducing a scaling parameter into the variational formula to achieve a better upper bound is a more conventional approach with a well-established history in generalization theory, such as PAC-Bayesian generalization bounds. For a detailed exploration of the importance of scaling parameters in upper-bounding techniques using $f$-divergence, we refer to [R1].
>
>
> [R1] Rohit Agrawal and Thibaut Horel. Optimal bounds between f-divergences and integral probability metrics. In ICML 2020.
>
>
>
> >- Additionally, clarify whether the claim in this paper—that there is no need to adjust the scaling parameter—holds generally.
>
> **Response.**  We note that we do theoretically justifies this in our Proposition 1 (simply because the surrogate loss $\hat{\ell}$ and $t$ are both unbounded). Specifically, in our Proposition 1, when the hypothesis space $\mathcal{H}$ is sufficiently large and the outputs of $\hat{\ell}\circ h'$ and $t\ell\circ h'$ span the entire space of $\mathbb{R}$ (as the crudest choice), then one expects that any optimal $\hat{\ell}\circ h'$ that maximizes $\tilde{d}$ can be matched by a corresponding $t{\ell}\circ h'$ to achieve the same value of $d$, and vice versa. This suggests that optimizing over $t{\ell}\circ h'$ is equivalent to optimizing over $\hat{\ell}\circ h'$. In practice, we do not know if $\mathcal{H}$ is large enough so that this argument holds, hence, we also empirically validate that optimization over $t$ is unnecessary in practice. See the Table 10 on Page 9 for details about optimizing over $t$.

---

> > ### Comment · Reviewer_vWvG · 2024-08-13
> >
> > Thanks for the clarifications on my concerns. Now I'm tend to the acceptance and raise my score.

---

> > > ### Author Response · Authors · 2024-08-13
> > > **Thanks**
> > >
> > > We are pleased to hear that your concerns have been addressed, and we sincerely appreciate your decision to raise the score.

---

### Official Review · Reviewer_EpWo · 2024-07-07

**Soundness:** 3
**Presentation:** 3
**Contribution:** 3
**Rating:** 7
**Confidence:** 2

**Summary:**

This paper studies the learning theory aspect of the domain adaptation problem, where the key is to bound the estimation errors between expectations over shifting distributions. Specifically, this work improves the recently developed $f$-divergence-based generalization analysis, where the main results ensure a tighter generalization upper bound and the consistency between theory and method. For finite sample setting, a sharp bound is provided to accelerate the asymptotic rate. Numerical simulation is conducted to demonstrate the superiority of the theory-guided method over the existing discrepancy-based framework.

**Strengths:**

+ The motivation is clear, i.e., improving the $f$-divergence-based bound and bridging the gap between method and theory, and the presentation is easy to follow.
+ The technical part is generally sound and the justifications are sufficient.
+ The experiment results are superior compared with recently developed generalization bounds.

**Weaknesses:**

+ Some notations are inconsistent in theoretical analysis.
+ The proposed algorithm needs further justifications.
+ The experiment comparison could be improved.

**Questions:**

There seem no major faults in this submission, and I only have the following minor concerns.

Q1. Theory and methodology. The major result for the target error bound is provided in Eq. (4) in Thm. 4.1 and the specific bound w.r.t. KL-divergence is presented in line 162, where the induced learning objective consists of source risk and the square root of cross-domain KL-divergence. However, it seems that the optimization objective Eq. (5) considers the divergence without the square root directly. I understand the optimal solutions are the same for these two objectives (if the optimal solutions ensure 0 cross-domain discrepancies). But considering Eq. (4) is closely related to the major merit of this work, i.e., the tight bound, the consistency between  Eq. (4)  and  Eq. (5) seems to be important. Some justifications are highly expected.

Q2. Method application. As far as I understand this work, the derived $f$-DD measure can be applied to existing works whose primary goal is discrepancy minimization. Thus, it could serve as a plug-and-play module for existing SOTA DA methods. Thus, some detailed discussions on the capability of $f$-DD w.r.t. existing methods are highly expected.

Q3. Following Q2, apart from the experiments in the current version, some comparisons between SOTA DA methods and their combination with $f$-DD objective are highly expected.

Q4. The clarity w.r.t. definitions could be improved, e.g., $K_{h',\mu}(t)$ depends on the hypothesis $h$ while the justification (i.e., line 178) is provided after the definition (i.e., line 176). A thorough check for these issues could improve the readability.

Q5. Some notations seem to be inconsistent. For example, the notation $I_{\nu}^{\phi}(h,h')$ in line 132 is inconsistent with $I$ in line 129; the notations $\mathbb{E}_{\nu}$ in line 132 seems to be incorrect (probably should be expectation over $\mu$?).

**Limitations:**

The limitations are discussed in the checklist, and there seems no potential negative societal impact.

---

> ### Author Rebuttal · Authors · 2024-08-06
>
> We thank you sincerely for your careful reading and valuable feedback on our paper. Our responses follow.
>
>
>
> >- The proposed algorithm needs further justifications.
>
> >- Q1. Theory and methodology. The major result ... the consistency between Eq. (4) and Eq. (5) seems to be important. Some justifications are highly expected.
>
> **Response.** The objective in the algorithm is in fact motivated by the findings from Section 5, where we demonstrate that when the error of the source domain is small, the $f$-DD term without the square-root function should more accurately reflect generalization (e.g., see Remark 5.3). Given that the empirical risk of the source domain is always minimized during training, we believe the final hypothesis lies within the subset of $\mathcal{H}$ (i.e., Rashomon set with $r_1$). Consequently, using Eq. (4) in this case is weak (as it has slow convergence). Note that $R^r_\mu(h)$ in Section 5 is upper-bounded by $r+r_1$, which we believe should not tend to infinity at the end of training.
>
> We will provide additional justification to clarify the proposed objective function in Eq. (5).
>
>
> >- The experiment comparison could be improved.
>
> >- Q2. Method application. As far as I understand this work, the derived $f$-DD measure can be applied to existing works whose primary goal is discrepancy minimization. Thus, it could serve as a plug-and-play module for existing SOTA DA methods. Thus, some detailed discussions on the capability of $f$-DD w.r.t. existing methods are highly expected.
>
> >- Q3. Following Q2, apart from the experiments in the current version, some comparisons between SOTA DA methods and their combination with $f$-DD objective are highly expected.
>
> **Response.** Thank you for your insightful suggestion regarding our $f$-DD being a plug-and-play module for existing SOTA methods. In fact, we do compare our $f$-DD with SOTA results on the Office-31 dataset, as shown in Table 10 of Appendix E. In particular, for a fair comparison, we replaced ResNet-50 with pretrained Vision Transformer (ViT) and Swin Transformer backbones. The results are close to the SOTA, with $93.9$ (ours) vs. $95$ (SOTA) for ViT and $94.8$ (ours) vs. $95.3$ (SOTA) for Swin-based transformer. In Lines 1214-1227, we provide a detailed explanation for why our $f$-DD currently does not outperform SOTA. Specifically, our $f$-DD lacks additional ingredients such as advanced pseudo-labeling techniques, Mixup, and label smoothing, which are commonly invoked in UDA algorithms.
>
> We acknowledge that integrating $f$-DD with SOTA methods could be beneficial, though it may require certain efforts for adjustments due to the complicated training objectives in these methods. Thus, rather than directly combining $f$-DD with SOTA methods, which could be complex, we have explored combining domain Mixup techniques with $f$-DD. Additionally, in response to other reviewers' requests for experiments on a larger dataset, we have included results on the VisDA-2017 dataset in the uploaded PDF (please find it in our general response). These results demonstrate the potential of $f$-DD to achieve SOTA performance.
>
> We will include these new results in the next revision.
>
>
> >- Some notations are inconsistent in theoretical analysis.
>
> >- Q4. The clarity w.r.t. definitions could be improved, e.g., $K_{h',\mu}(t)$ depends on the hypothesis $h$ while the justification (i.e., line 178) is provided after the definition (i.e., line 176). A thorough check for these issues could improve the readability.
>
> **Response.** Thanks for pointing out this. We will revise the definition of $K_{h',\mu}(t)$, and thoroughly proofread the paper to identify any similar issues.
>
> >- Q5. Some notations seem to be inconsistent. For example, the notation $I^\phi_{\nu}(h,h')$ in line 132 is inconsistent with $I$ in line 129; the notations $\mathbb{E}_{\nu}$ in line 132 seems to be incorrect (probably should be expectation over $\mu$?).
>
> **Response.** Thank you for catching that. You are correct—these are typos. Specifically, the $I$ in line 129 should be $I^\phi_{\nu}(h,h')$, and the expectation should be with respect to $\mu$. We appreciate your careful reading.

---

> > ### Comment · Reviewer_EpWo · 2024-08-11
> >
> > I thank the authors for providing responses to the concerns. My concerns are well-addressed.
> >
> > Overall, I believe that this paper provides a tighter generalization error analysis framework by exploring a new 'change of measure' inequality via $f$-divergence, which is generally better than the commonly employed divergence based on the integral probability metric. Thus, I would like to improve my score accordingly.

---

> > > ### Author Response · Authors · 2024-08-12
> > > **Thank you**
> > >
> > > We sincerely appreciate the reviewer’s positive feedback on our response and are grateful for the increased score.

---

### Official Review · Reviewer_CqSQ · 2024-07-09

**Soundness:** 3
**Presentation:** 3
**Contribution:** 2
**Rating:** 6
**Confidence:** 3

**Summary:**

This paper aims to develop an improved version of f-divergence-based unsupervised domain adaptation (UDA) learning theory. In particular, the authors introduce a novel f-divergence-based domain discrepancy measure (f-DD) by combining the two existing concepts, which are f-divergence and domain discrepancy. Based on that f-DD measure, the paper next provides a generalization bound on the target domain, which is shown to be sharper than the existing related bound. The experimental results consistently demonstrate that f-DD outperforms the original f-DAL in three popular UDA benchmarks, with the best performance achieved by Jeffereys-DD.

**Strengths:**

The paper is well-written and easy to follow. The idea of introducing f-divergence-based UDA, targeting a better risk-bound on the target domain is novel and interesting. All the main statements of the paper are theoretically supported, though I did not have enough time to verify all of those propositions/theorems carefully.

The experimental results consistently demonstrate that f-DD outperforms the original f-DAL in three popular UDA benchmarks, with the best performance achieved by Jeffereys-DD.

**Weaknesses:**

The novelty of the paper is quite limited since the f-divergence-based domain discrepancy measure (f-DD) is proposed by combining the two existing concepts, which are f-divergence and domain discrepancy.

In Theorem 5.2, the authors claim that the application of the localization technique gives a fast-rate generalization, they do not provide a concrete evidence. Could the author give some explanations/clarifications for that.

Moreover, the experimental part of the paper seems not be very convincing since it only provides experiments with quite small datasets (Office31, Office-Home, MNIST & USPS) and simple model (e.g., Lenet). It raises the concern about capability of f-DD in more complicated settings with large datasets and backbone network.

**Questions:**

Please refer to my comments about the weaknesses of the paper.

---

> ### Author Rebuttal · Authors · 2024-08-06
>
> We thank you sincerely for your constructive comments. Our responses follow.
>
> >- The novelty of the paper is quite limited since the f-divergence-based domain discrepancy measure (f-DD) is proposed by combining the two existing concepts, which are f-divergence and domain discrepancy.
>
> **Response.** We respectfully disagree with the assessment that the novelty of our paper is limited due to the combination of $f$-divergence and domain discrepancy. Our paper aims to advance the existing framework rather than merely combining these concepts. The title of our paper highlights that we present an "improved framework" compared to the existing work (proposed by Acuna et al. (2021)), reflecting our focus on improving the previous results. While the variational representation of $f$-divergence is indeed discussed in earlier literature, the novel aspect of our paper lies in its application within a new $f$-divergence-principled theory for UDA.
>
> We now summarize our novel theoretical contributions here: 1) Acuna et al. (2021) introduced the absolute value function into their discrepancy definition, leading to a disconnection from the original $f$-divergence, and potentially allowing it to go to infinity (as illustrated in Figure 1 in our paper). Their Lemma 1 claims that their notion is upper-bounded by $f$-divergence; but unfortunately their Lemma 1 is flawed. Specifically, their Eq. (B.5) in the appendix is incorrect as they use $\sup A = \sup |A|$ whereas essentially $\sup A \leq \sup |A|$, and hence the equality is not justified. This disconnection suggests that, strictly speaking, a rigorous $f$-divergence-based theoretical guarantee for the target error in UDA has not been established prior to our work; 2) In addition, we apply the localization technique in the analysis. The localization technique has been studied extensively in localized Rademacher complexity since 2005, or even earlier in the context of empirical processes theory. In UDA, before Zhang et al. (2020), the localization technique was mentioned in Example 4 of Hanneke & Kpotufe (2019). The novelty in our Section 5 lies in our novel application of localization: previous works like Zhang et al. (2020) utilize localization to achieve better sample complexity results  (e.g., $\mathcal{O}(\frac{1}{n}+\frac{1}{m})$), while we directly incorporate the localization technique prior to proving the sample complexity results. In Lemma 5.1, we use it to remove the square-root function in Eq. (4) from Section 4, and as expected, it also helps to improve sample complexity in Theorem 5.2. Our proof techniques are in fact quite novel.
>
>
> We sincerely hope the reviewer could reconsider the novelty and impact of our contributions in light of these clarifications.
>
> >- In Theorem 5.2, the authors claim that the application of the localization technique gives a fast-rate generalization, they do not provide a concrete evidence. Could the author give some explanations/clarifications for that.
>
> **Response.** We would like to direct the reviewer to Appendix D.12, as also referenced in Line 259 of the main text, where we provide a concrete example demonstrating this. Specifically, in the threshold learning example discussed in Appendix D.12, we show through straightforward calculations that the quantity
> $5*\mathrm{D}^{h\_{\frac{1}{2}},\mathcal{H}\_{\frac{1}{4}}}\_{\rm KL}(\nu||\mu)+0.1*R^r_\mu=0.265$ is indeed smaller than $\sqrt{\mathrm{D}^{h\_{\frac{1}{2}},\mathcal{H}}\_{\rm KL}(\nu||\mu)}=0.36$. This illustrates that the localization technique results in a tighter bound.
>
> >- Moreover, the experimental part of the paper seems not be very convincing since it only provides experiments with quite small datasets (Office31, Office-Home, MNIST & USPS) and simple model (e.g., Lenet). It raises the concern about capability of f-DD in more complicated settings with large datasets and backbone network.
>
> **Response.** For additional experimental results of more complicated networks, please refer to Lines 1201-1227 in Appendix E, where we present results using the pretrained Vision Transformer (ViT) base model and the pretrained Swin Transformer. Additionally, ResNet-50 was used for the Office31 and Office-Home datasets, while LeNet was used only for the Digits dataset, where it has already demonstrated strong performance.
>
> Furthermore, we have included experimental results on the VisDA-2017 dataset in the uploaded PDF file (please see our general response).

---

> > ### Comment · Reviewer_CqSQ · 2024-08-12
> > **Upgrading my score to 6**
> >
> > Thanks to the authors for the detailed responses, especially for clarifying the scope of the paper. Since most of my concerns have been addressed, I upgraded the paper's score to 6.

---

> > > ### Author Response · Authors · 2024-08-12
> > > **Thank you**
> > >
> > > We are glad the reviewer is satisfied with our response, and we really appreciate the improved score.

---

### Official Review · Reviewer_HkGQ · 2024-07-23

**Soundness:** 3
**Presentation:** 2
**Contribution:** 4
**Rating:** 4
**Confidence:** 3

**Summary:**

This paper improves the theoretical foundations of UDA proposed by previous work, named f-DD. By removing the absolute value function and incorporating a scaling parameter, f-DD yields novel target error and sample complexity bounds, allowing us to recover previous KL-based results and bridging the gap between algorithms and theory presented in Acuna et al. Leveraging a localization technique, this paper also develops a fast-rate generalization bound. Empirical results demonstrate the superior performance of f-DD-based domain learning algorithms over previous works in popular UDA benchmarks.

**Strengths:**

1) This paper holds significant theoretical significance in the field of UDA (Unsupervised Domain Adaptation);
2) The proof of the theorem is very solid;
3) The experiments are also sufficient.

**Weaknesses:**

1) The readability of the paper is poor. It is almost entirely composed of definitions, remarks, lemmas and theorems, lacking a figure to introduce the motivation of this paper and explain why the improved framework is effective. 2) It is difficult to reproduce the results, as the training objective (5) is very abstract and unclear how to implement it experimentally. 3) This paper requires a substantial foundation of reading other papers in order to be understood.

**Questions:**

How to implement the training objective (5)?

**Limitations:**

Please refer to the weakness.

---

> ### Author Rebuttal · Authors · 2024-08-06
>
> We thank you sincerely for your valuable feedback on our paper. Our responses follow.
>
>
> >- The readability of the paper is poor. It is almost entirely composed of definitions, remarks, lemmas and theorems, lacking a figure to introduce the motivation of this paper and explain why the improved framework is effective.
>
> **Response.** Thank you for highlighting this concern. We understand that the paper may seem heavy on definitions, remarks, lemmas, and theorems, as it falls under the "Learning Theory" category, where theoretical analysis is often central.
>
> We appreciate your suggestion to include a figure to improve readability. We have added a figure in the updated PDF that provides an overview of our improved framework (please see our general response). Additionally, we note that our improved framework is more effective because it provides a better variational representation of $f$-divergence compared to previous approaches. This, being one of our key motivations, is inevitably supported by a theoretical perspective.
>
> >- It is difficult to reproduce the results, as the training objective (5) is very abstract and unclear how to implement it experimentally.
>
> >- How to implement the training objective (5)?
>
> **Response.** We hope the uploaded figure in the general response could help to clarify the implementation of training objective. Specifically, it is implemented similarly to the adversarial training strategy proposed in DANN [R1]. To restate, our training objective is:
> $$ \min_{h\in\mathcal{H}}\max_{h'\in\mathcal{H'}}R_{\hat{\mu}}(h)+ \eta {d}_{\hat{\mu},\hat{\nu}}(h,h').
> $$
>
> Our model is a deep neural network (e.g., ResNet-50) consisting of a representation network (i.e., $h_{\text{rep}}$) and two classification networks. The main classification network (i.e., $h\_{\text{cls}}$) is used for predictions, while the auxiliary classification network (i.e., $h'_{\text{cls}}$) is used to calculate the domain disagreement (e.g., $\ell(h\_{\text{rep}} \circ h\_{\text{cls}}, h\_{\text{rep}} \circ h'\_{\text{cls}})$).
>
> For the outer minimization, $\min\_{h} R\_{\hat{\mu}}(h) + \eta \max\_{h'} {d}\_{\hat{\mu},\hat{\nu}}(h, h')$, we fix the parameters in the auxiliary classification network and jointly minimize the classification loss for the source domain data (i.e., $R\_{\hat{\mu}}(h)$) and the approximated $f$-DD between the source and target domains (i.e., $\max\_{h'} {d}\_{\hat{\mu},\hat{\nu}}(h, h')$). Note that $\max\_{h'} {d}\_{\hat{\mu},\hat{\nu}}(h, h')$ represents the empirical version of our $f$-DD. Then,
> for the inner maximization, $\max\_{h'} {d}\_{\hat{\mu},\hat{\nu}}(h, h')$, we fix the main classification network $h\_{\rm cls}$ and then maximize ${d}\_{\hat{\mu},\hat{\nu}}(h, h')$ by training the auxiliary classification network.
>
> This adversarial training strategy effectively minimizes both the source domain risk and the domain discrepancy between the source and target domains. We will provide additional details on the implementation to ensure completeness.
>
> [R1] Yaroslav Ganin, et al. "Domain-adversarial training of neural networks." Journal of Machine Learning Research 17.59 (2016): 1-35.
>
> >- This paper requires a substantial foundation of reading other papers in order to be understood.
>
> **Response.**  We acknowledge that understanding this paper may require some background knowledge. However, we have made efforts to self-contain all theoretical background within the main text or the Appendix to make it more friendly to a broader audience. We would appreciate it if  the reviewer could point out more specific areas that may be unclear or less accessible to general readers. We are more than willing to improve the presentation of our paper and welcome any constructive feedback that can help enhance its clarity. We sincerely hope that requiring comprehensive background knowledge is not taken as a negative point against the acceptance of this paper.

---

### Official Review · Reviewer_zRKu · 2024-07-25

**Soundness:** 3
**Presentation:** 3
**Contribution:** 3
**Rating:** 7
**Confidence:** 3

**Summary:**

In this paper, new expected risk analysis based on f-divergence is provided for the unsupervised domain adaptation problem. Although there are prior researches on expected risk analysis based on f-divergence, several issues have been pointed out, such as the fact that the variational representation of f-divergence used in these studies does not recover the Donsker-Varadhan representation of KL-divergence, and the use of the absolute value of the variational representation as a measure of domain discrepancy.
In this paper, to address these issues, the authors adopt an alternative variational representation of f-divergence and, based on this, provide an upper bound evaluation of the expected risk in the target domain, namely ``target risk $\le$ source risk + marginal domain discrepancy + joint error''. Additionally, a sample approximation version of the derived upper bound is also provided, allowing it to be estimated from the data  (excluding the joint error part, as in conventional bounds).

**Strengths:**

- The paper clearly discusses what are difficulties  with the conventional DA theory using f-divergence and explains how it is solved by the proposed approach. Especially, this paper provides a solid theoretical foundation, with detailed assumptions and rigorous proofs that are well-documented in the appendix.

- Previous expected risk bounds in UDA have often been given by relatively simple inequality evaluations, following the formulation given by Ben-David et al. In contrast, a similar upper bound evaluation using the f-DD proposed in this paper requires an inequality evaluation for ``change of measure" (as given in Lemma 4.1), and it can be seen that this is not an incremental extension of the conventional DA theory.

**Weaknesses:**

- I don't think there is enough information needed when trying to calculate the derived upper boundary from the sample. For example, $t_0$ in Lemma 4.2 and the construction of the Rashomon set in Sec 5 should be discussed in more detail.

**Questions:**

- Are no assumptions specifically placed on the hypothetical set $\mathcal{H}$?

- In Lemma 4.2, is there a way to estimate the value of t_0 when t_0 cannot be written in closed form (e.g. for KL or Jeffereys employed in the Experiments?)?

- The Rashomon set used in Section 5 appears to be in fact the set to be estimated (as the true expected risk of the source domain is unknown). How exactly is the Rashomon set constructed in this paper?

- In the experiments, three types of discrepancies are evaluated for the proposed method, namely KL-DD, $\chi^2$-DD and Jeffereys-DD. Then, do you have any insights on which discrepancy measure should be used for which type of problem? I think the question of which measure to use to evaluate domain discrepancy is critical not only for theorists but also for practitioners.

- Is the f-DD proposed in Definition 4.1 always 'better' than the existing f-divergence-based discrepancy (Definition 3.1)? I am wondering whether there are cases where using absolute values to define the discrepancy is an advantage.

**Limitations:**

The authors have adequately addressed the limitations and potential negative societal impact of their work.

---

> ### Author Rebuttal · Authors · 2024-08-06
>
> We thank you sincerely for your positive evaluation and constructive comments. Our responses follow.
>
> >- Are no assumptions ...?
>
> **Response.** Our theoretical results, except for Proposition 1 which requires $\mathcal{H}$ to be sufficiently large, do not rely on any specific assumptions about $\mathcal{H}$. The hypothesis class can be either finite or infinite. Additionally, in our localized $f$-DD, we restrict the entire hypothesis class to a smaller Rashomon set. While we do not require any assumptions on $\mathcal{H}$ here, we focus our analysis on the relevant hypotheses, as those that perform poorly on the source domain are unlikely to be found by any effective UDA algorithm.
>
> >- In Lemma 4.2, is there a way to estimate ...?
>
> **Response.** We have discussed the method for approximating $t_0$ (or the optimal $t$ equivalently) in Lines 339-354 (and Line 1183-1195), which is based on a quadratic approximation, and compared the results with simply setting $t_0=1$ in Table 4. Moreover, there are other methods to estimate $t_0$ as well. For example, assigning $t_0$ an initial value and considering it a trainable parameter, then applying SGD to update it during training.
>
> We will highlight the estimation of $t_0$ in a more noticeable way in our next revision.
>
> >- The Rashomon set used ... How exactly is the Rashomon set constructed in this paper?
>
> **Response.**  Please notice that in our experiments, we do not need to explicitly construct the Rashomon set or determine its threshold value. The hypothesis found by UDA algorithms is always within some Rashomon set of our interest. In practical scenarios, where the true risk of the source domain is unknown, its empirical risk serves as a proxy for the true risk. As long as the UDA algorithm minimizes the classification error of the source domain, the obtained hypothesis falls within a restricted hypothesis class, i.e., the Rashomon set with some relatively low threshold value $r$, although the exact value of $r$ may be unknown as it relates to the generalization error of the source domain.
>
> We believe the confusion might stem from the motivation behind our localized $f$-DD. We apply localization techniques to show that the convergence rate based on $f$-DD could be faster if we restrict the entire hypothesis class to the Rashomon set. This approach aligns with practical applications because, for any UDA algorithm, regardless of how the source and target domains are aligned, a good classifier must first be trained on the source domain through empirical risk minimization. Given that the hypothesis should first perform well on the source domain, many hypotheses are not of interest in theoretical analysis. By removing such redundancy, the UDA algorithm enjoys a fast-rate generalization guarantee (cf. Theorem 5.2). Consequently, the training objective in our algorithm, namely Eq. (5), does not need to include a square-root function for $f$-DD, in contrast to Eq. (4), which considers the entire hypothesis class.
>
> Additionally, to demonstrate the advantage of localized $f$-DD, we provide a threshold learning example in Appendix D.12, where the Rashomon set can be explicitly constructed since the ground truth hypothesis is known.
>
> >- In the experiments, three types of discrepancies are ...
>
> **Response.** When comparing KL-DD and our weighted Jeffreys-DD, we always recommend using the weighted Jeffreys-DD. This measure is a weighted combination of KL and reverse KL (i.e. $\gamma_1 \mathrm{D}\_{\rm KL}(\hat{\mu} \|\ \hat{\nu}) + \gamma_2 \mathrm{D}\_{\rm KL}(\hat{\nu} \|\ \hat{\mu})$). In most UDA problems, the difficulty of transferring knowledge from the source domain to the target domain may not be the same as transferring from the target domain to the source domain. From this perspective, the weighted Jeffreys-DD provides more flexibility to reflect this difference during training.
>
> When comparing KL-DD and $\chi^2$-DD, while KL is theoretically smaller than $\chi^2$, their empirical performance is quite similar. In fact, $\chi^2$-DD may even give better results on some sub-tasks, such as A$\rightarrow$W in Office-31. We believe there may be some practical optimization benefits of $\chi^2$ on certain sub-tasks (e.g., large quantity being more informative), which might not be easily explained by generalization bounds.
>
> Overall, we believe that weighted Jeffreys-DD is a good option in practice, especially when no further prior knowledge of the dataset is available.
>
> >- Is the f-DD proposed in Definition 4.1 always 'better' ...
>
> **Response.** In the context of approximating the true $f$-divergence between two distributions, our $f$-DD is indeed always better than the existing $f$-divergence-based discrepancy (Definition 3.1). This is because the additional absolute value function transforms all negative elements to positive, leading to a larger and more pessimistic approximation. Beyond the scope of approximating true $f$-divergence, while we are not aware of any hypothesis classes where Definition 3.1 is smaller than $f$-DD, from our current understanding, such cases are unlikely to be of practical interest. We are, however, open to discussing this further. Additionally, even without the absolute value function, the previous $f$-DAL defined in Acuna et al. (2021) is based on a weaker variational representation of $f$-divergence. As a lower bound of $f$-divergence, our $f$-DD is always better to theirs, as detailed in Appendix C.4.
>
> Furthermore, it is worth noting that while the $f$-DAL paper proposes an absolute value $f$-divergence-based discrepancy, it removes the absolute value function in their algorithm. Thus, the practical impact of the absolute value function has not been demonstrated in any existing works.

---

### Author Rebuttal · Authors · 2024-08-06

We would like to thank all the reviewers for their constructive comments and valuable feedback. In addition to addressing your individual comments separately, we have also uploaded a PDF file that contains a figure and a table. Specifically:

1. Figure: In response to Reviewer HkGQ's comment that the paper lacks a figure to illustrate the proposed method and that the implementation of the objective function is not straightforward, we have added an overview figure, which introduces the adversarial training framework of our $f$-DD-based UDA algorithm.


2. Table: Reflecting suggestions for additional experimental results: Reviewer CqSQ recommended conducting experiments on a larger dataset and with a more complex network structure, Reviewer EpWo suggested combining our $f$-DD with SOTA UDA methods and Reviewer vWvG also suggested testing $f$-DD on additional datasets and tasks.

To accommodate these suggestions, we have included additional results on the VisDA-2017 dataset in the table, which is more challenging than those used in our original paper. In particular, we replaced ResNet-50 with the Swin Transformer as the backbone, and we also attempted to apply domain mixup, a crucial component in many existing SOTA methods, into our $f$-DD algorithm. Specifically, we mixed source and target domains at the pixel level to create a third mixed domain, then jointly minimized the $f$-DD between the source and mixed domain and the $f$-DD between the target and mixed domain. This method improved performance, bringing it close to the SOTA method, e.g., PMTrans.

Due to time constraints during the rebuttal period, we trained $f$-DD for only 20 epochs, which is relatively few compared to SOTA methods. We expect that with additional effort in tuning hyperparameters, the performance of $f$-DD (with and without MixUp) can be further boosted.

We will include the figure and the new experimental results in the revision.

---

### Decision · Program_Chairs · 2024-09-25

**Decision:**

Accept (poster)

**Comment:**

This paper proposes to improve the existing framework for unsupervised domain adaptation based on the notion of f-divergence proposed by Acuna et al.(2021) by defining new f-domain discrepancy leading to novel and improved bounds on the target error. Using a localization technique, some fast rate generalization bound is also provided.

During rebuttal authors have answered to numerous issues raised by the reviewers, including adding some clarification on the contribution and its scope, adding a figure to summarize the principle of the approach and discussing the improvement of the empirical results for comparing SOTA results with a new experiment on the VisDA dataset.
Overall, reviewers have appreciated the contribution notably the fact to offer tighter generalization bounds with a new change of measure inequality via f-divergence, a novel localization approach with original proof, and the potentiality to reach SOTA results while providing a strong theoretical framework.

I propose then to accept the paper.
I strongly recommend that authors take into account the feedback of the reviewers in the final version, in particular for the correction of the typos, and the integration of the different explanations and clarifications provided in the rebuttal (including the proposed figure) as well as the additional experiments.